# THEORY OF LLM SAMPLING: PART DESCRIPTIVE AND PART PRESCRIPTIVE

## ABSTRACT

Large Language Models (LLMs) are increasingly utilized in autonomous decision-making systems, where they sample options from an action space. However, the underlying heuristics guiding the sampling of LLMs remain under-explored. We examine LLMs' response sampling and propose a theory that the sample of an LLM is driven by a descriptive component (the notion of statistical average) and a prescriptive component (notion of an ideal represented in the LLM). In a controlled experimental setting, we demonstrate that LLMs' outputs deviate from the statistically probable outcome in the direction of a prescriptive component. We further show that this deviation towards a prescriptive component consistently appears across diverse real-world domains, including social, public health, and scientific contexts. Using this theory, we demonstrate that concept prototypes in LLMs are affected by prescriptive norms, similar to the concept of normality in humans. Through case studies, we illustrate that in real-world applications, the shift toward an ideal value in LLMs' outputs can result in significantly biased decision-making, raising ethical and trustworthiness concerns.

## 1 INTRODUCTION

LLMs are often considered to be 'System-1'(Daniel, 2017), characterized by their reliance on heuristics, operating implicitly without deliberative reasoning (Dasgupta et al., 2022; Yao et al., 2023). Their performance in embodied decision making Li et al. (2024), expansive action spaces Wen et al., planning in action spaces Valmeekam et al. is attributed to the heuristics and mechanisms driving their operation. While LLMs are benchmarked as autonomous decision-making systems sampling options from an action space, the underlying heuristics guiding their response sampling remain under-explored.

We study this heuristics and propose a theory that the sampling of an LLM is driven by a descriptive norm (the notion of statistical average) and a prescriptive norm (a notion of an ideal represented in the LLM)(Figure 1). We define response sampling as the process by which the model probabilistically selects outputs from a distribution of potential responses. A descriptive component represents what is observed or statistically likely within a given context, reflecting the occurrence or probability of observations without implying any value judgment. A prescriptive component is an implicit standard of what is considered ideal, desirable, or valued within on a concept, often encoded by grades/scores that prioritizes outcomes deemed "better/optimal". The proposed theory implies, the sample of an LLM not only reflects the statistical regularities of the data (descriptive norms) but also systematically incorporates an idealized version of the concept (prescriptive norms).

We design a critical experiment to validate the proposed theory. We show that the effect of this heuristics appears consistently across diverse real-world domains. We perform extensive experiments covering different LLMs, evaluated concepts, and ablations to show the robustness of observations. We present a case study where an LLM is used to predict medical recovery time of patients to show a practical implication of the LLM having a prescriptive component in sampling. To explain the theory, we rely on its convergence with how humans consider options. Heuristics employed by humans is driven by concept prototypicality which has a prescriptive component (e.g., a prototypical teacher is one that teaches well). In short, we make the following contributions:

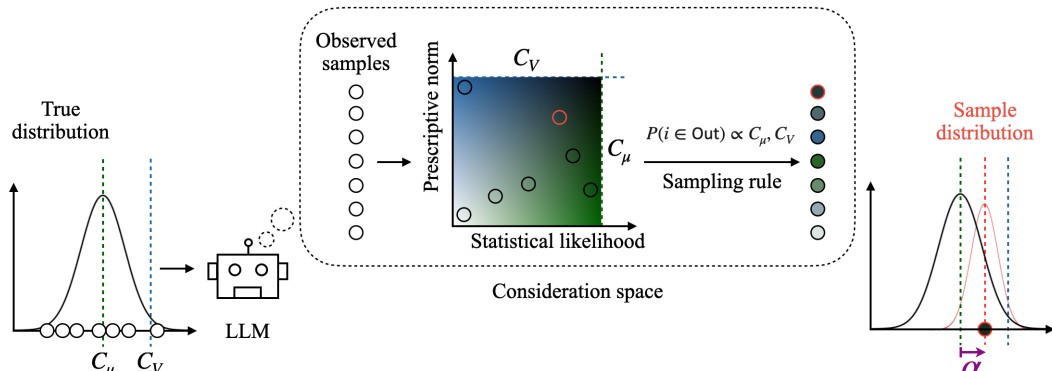

Figure 1: When sampling from a set of options, the LLM selects the sample that is both statistically likely and closely aligned with a prescriptive ideal. Consequently, the sampled distribution exhibits a shift ($\alpha$) away from the true distribution in the direction of the ideal.

- We study the sampling mechanisms in LLMs through the lens of cognitive studies in humans, and show that the heuristics driving the sampling processes of both humans and LLMs converge on having a descriptive component and a prescriptive component. We construct three major experimental settings to empirically validate the proposed theory with many robustness checks.

- We evaluate samples from a range of 500 existing concepts across 10 domains to verify the validity of the proposed theory and find the results, on 15 language models covering different families and sizes, to be statistically significant. We show a case study inspired by real-world applications where this prescriptive component may lead to undesired outcomes.

- We study the proposed theory on concept prototypicality. We also show that the ideal notion in LLMs might not align with the value system of humans even though both LLMs and humans seem to share the same heuristic components.

## 2 RELATED WORK

**Understanding LLMs as 'System-1':** Reasoning has been broadly characterized as a two-step process involving quick 'System-1' thinking and a more deliberate 'System-2' reasoning (Daniel, 2017). Large Language Models (LLMs) have been conceptually likened to System-1 reasoning due to their automatic and implicit nature (Yao et al., 2023). In fact, recent studies show overlaps in errors made by LLMs and humans in System-1 reasoning tasks, indicating that both might rely on heuristics for rapid decision-making (Dasgupta et al., 2022). We systematically study the heuristics that drive sampling in LLMs.

**Understanding heuristics in response sampling:** Simon (1996) uses the notion of heuristics to explain the decision-making of 'System-1' mechanisms. These studies demonstrate the utility of 'mental shortcuts' to navigate countless possibilities of the search problem (Newell et al., 1972). In the case of LLMs, exploring their heuristics can offer insights into how these models process information. However, previous research mainly uses sampling for tasks like action generation and decision-making rather than to explicitly understand the internal heuristics at work in LLMs (Hazra et al., 2023; Shah et al., 2023; Suri et al., 2023). Our work aims to fill this gap by investigating the heuristics driving LLM response sampling, which could provide a deeper understanding of their decision-making processes.

Earlier work that examined the mechanisms by which LLMs generate outputs highlights that LLMs may produce coherent text by probabilistically assembling language patterns without 'genuine understanding' Bender et al. (2021). Later investigations have demonstrated that LLMs can develop internal, structured representations of the environment Li et al. (2022), and when trained on programming languages exhibit an understanding of semantic structures, indicating a capacity for meaningful text processing and generationJin & Rinard (2023). This slightly contradicting views on interpreting LLM outputs shows the significance of further explorations. Recent work indicates that LLM agents can understand probabilities, but they struggle with probability sampling Gu et al. (2024), hindering their effectiveness in generating samples that align with expected probabilistic patterns. Our paper provides a systematic framework that explains the sampling behaviour of LLMs. It enables precise exploration of LLM decision-making heuristics across diverse domains.

## 3 THEORY OF LLM SAMPLING

Prominent theories explain decision-making in humans and animals as a search problem of countless possibilities (Phillips et al., 2019; Phillips & Cushman, 2017; Mattar & Lengyel, 2022; Ross et al., 2023). To navigate a huge search space of possibilities, Simon (1996) propose that humans (as well as machines) must rely on heuristics to simplify the decision-making process (Newell et al., 1972). Increasing evidence shows that humans use likelihood and value as heuristics (Bear et al., 2020; Phillips et al., 2019; Bear & Knobe, 2017b). This dual nature of thought is hypothesized to originate from humans being goal-driven agents and engaging in value maximization (Bear & Knobe, 2017b).

This human possibility sampling follows two stages: first, a fast but less accurate, heuristic driven system generates a set of reasonable options (System-1), followed by a deliberate, but more precise, system that selects the best choice (System-2) (Phillips et al., 2019). It is the heuristics of the first stage that enable humans to make quick and effective decisions. LLMs are understood as System-1 machines that are driven by heuristics (Yao et al., 2023). In light of these studies, we examine the sampling mechanisms of LLMs and observe that both LLMs and humans converge on the same heuristics as the sampling is driven by the average and the ideal. Based on this, we propose a theory for LLM sampling:

> The sampling of an LLM is driven by a descriptive component (the notion of statistical average) and a prescriptive component (a notion of an ideal).

Here, sampling is defined as the process by which the model probabilistically selects outputs from a distribution of potential responses. In the following subsections, we describe the empirical evaluation of this theory.

### 3.1 SAMPLING IN RELATION TO A NOVEL CONCEPT

To empirically validate the proposed theory, we construct a setting by introducing a novel concept $C$. This approach eliminates potential confounding effects associated with using pre-existing concepts. We present the LLM with a task to sample values from a range of possibilities on this concept and evaluate the samples to uncover the effect of prescriptive and descriptive norms on sampling.

To establish a statistical baseline for the concept, we construct a distribution for concept $C$ with a mean $C_\mu$. The LLM is provided with $N$ samples from this distribution as values associated with $C$. We denote these options observed by the LLM as $C_o$. In our experiments, we chose a sufficiently large $N$, such that the mean of observed options is almost equal to $C_\mu$. To establish a prescriptive norm $C_v$ on the concept $C$, we associate each option $C_o$ with a prescriptive component, represented by a grade. The grade associated with each observed sample gives a prescriptive norm to the concept; we repeat the experiment with the following conditions: a higher value being ideal, a lower value being ideal, and a control experiment. Based on these inputs (the observed $N$ samples along with the corresponding grades if any), we prompt the LLM to sample a value for the concept $C$. We denote the value sampled by the LLM as $C_s$. By changing $C_\mu$ and $C_v$ and keeping the rest of the prompt the same, we show how the value of $C_s$ changes with these two components.

In independent contexts (i.e., prompts), we repeat this procedure $M$ times to obtain a sample distribution. We keep the value of $M$ the same as $N$ in all variants of the experiment. We evaluate whether the distribution of samples $C_s$ generated by the LLM is significantly different from the distribution of input samples $C_o$. If the sample is driven solely by the descriptive norm (statistics of the observed samples), the distribution of samples $C_s$ is expected to be statistically similar to the observed distribution. However, the difference between observed samples and output samples might occur due to the error in approximating the statistics of the observed samples. To exclude this possibility, we instruct the LLM to report the average of the distribution. We denote the reported average by $C_a$. Across all experiments, we observe that $C_\mu \approx C_a$, indicating that the LLM reliably approximates the statistics of the observed distribution. The control run also helps validate this.

We apply the Mann-Whitney U test to compare the distribution of samples, $C_s$, with the average reported by the LLM, $C_a$, and the true mean of the observed samples, $C_\mu$. We vary the direction of $C_v$ and demonstrate that the change in samples' mean (mean of $C_s$) corresponds to the change in $C_v$. For each concept, $C$, we calculate the Mann-Whitney U statistic and the corresponding $p$-value. If

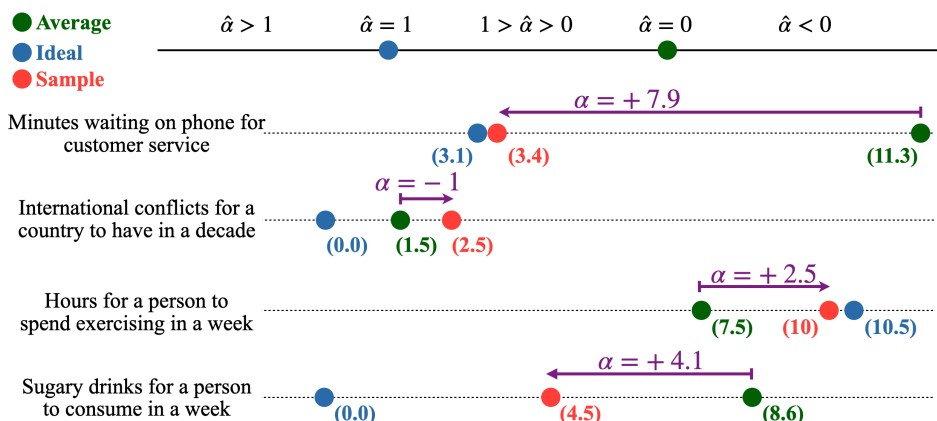

Figure 2: The figure shows the average, ideal, and sample values reported by the LLM for four different concepts. Positive $\alpha$ shows the deviation in the direction of the ideal. The sample is not just following the notion of average, but is also driven by ideal.

$p < 0.05$, there is a significant difference between the distribution of $C_s$ and $C_a$. Then, we check if the shift corresponds to the direction of the prescriptive component.

## 3.2 SAMPLING IN RELATION TO EXISTING CONCEPTS

We investigate the validity of the theory outside the constructed setting using existing concepts in the LLM. We test the theory on multiple concepts learned during pre-training across different domains. Here, the distribution and the prescriptive norm are unknown. Therefore, we ask the LLM to report $C_a$ (the average) and $C_i$ (the ideal), and then to pick a sample, $C_s$. Note that $C_v$ and $C_\mu$ are not known in this scenario. We use a binomial test to determine whether the sample $C_s$ falls on the ideal side of the average or the non-ideal side of the average. The latter can also be understood as the sample falling on the average side of ideal. We classify each sample $C_s$ as:

$$\text{Ideal side of average} : \begin{cases} C_s > C_a & \text{if } C_i > C_a \\ C_s < C_a & \text{if } C_i < C_a \end{cases} \tag{1}$$

$$\text{Average side of ideal} : \begin{cases} C_s < C_a & \text{if } C_i \geq C_a \\ C_s > C_a & \text{if } C_i \leq C_a \end{cases} \tag{2}$$

Samples of both concepts are shown in Figure 2. Consider the number of concepts for which sample falls on the ideal side of the average is $n$ and the total number of concepts evaluated is $n_{total}$. The binomial test is used to determine if $n$ is significantly different from what would be expected by chance, assuming a null hypothesis where the probability $p$ of a sample being on the ideal side is 0.5. The $p$-value obtained from the binomial test is used to assess significance. $p < 0.05$ shows a significant presence of prescriptive norm across concepts.

The setting described in Sections 3.2 and 3.1 is inspired by similar evaluation in humans (Bear et al., 2020; Phillips et al., 2019; Bear & Knobe, 2017b). We scale the experiments to show higher statistical significance and later replicate the exact setting to compare results with human studies.

**Drift from the statistical norm:** In most applications, one might expect the LLM sampling to be driven by the statistical likelihood alone. We use a variable $\alpha$ to quantify the degree to which the sample deviates away from the statistical norm. We define $\alpha$ such that, when the proposed theory holds, the value of $\alpha$ is positive. That is, $\alpha$ is measured to be positive when $C_s$ deviates from the $C_a$ in the direction of $C_i$. We compute this direction as the positive direction of $\alpha$ (Figure 2). For each sample $C_s$ of a concept $C$, $\alpha$ is computed as

$$\alpha = (C_a - C_s) \times sign(C_a - C_i) \tag{3}$$

We also compute $\hat{\alpha}$: a normalized scale such that $C_a$ is at the origin and $C_i$ is at unit distance from the origin. We compute $\hat{\alpha}$ as $\alpha/|C_a - C_i|$. $\hat{\alpha}$ enables comparison across concepts with less dependency on the scale of values. It also allows comparison with observations obtained in the experiments with human subjects.

### 3.3 Prescriptive component in concept prototypes

A prototype is the most typical or representative member of a concept, often viewed as the "average example" based on shared features or frequency of occurrence (Murphy, 2004). But, it also serves as a mental benchmark, embodying both statistical regularities and goal-oriented ideals within a concept (Barsalou, 1985). For instance, a 'Robin' might be considered a prototype of the concept 'Bird', as it shares many common features with most birds with high occurrence, and has the ability to fly (expected of birds), making it a representative example of the 'concept' (Smith & Medin, 1981). In this way, prototypicality can be used to understand the normality of a concept 1.15.

Our aim is to determine whether the LLM's judgment of prototypes is influenced solely by statistical regularities or whether prescriptive (goal-oriented) ideals also play a role. We provide a concept $C$ and corresponding exemplars of that concept. We ask the LLM to judge on three dimensions, namely the average, ideal, and the prototypicality of the exemplar. As in the previous section 3.2, we check whether the prototypicality rating falls on the ideal side of the average. To test significance, we do a binomial test across concepts $C$ to check if LLMs conception of prototypes has a perspective component. The evaluation is similar to the previous section.

## 4 Experiments and Results

In this section, we present three key experiments. First, we present a constrained setting to test the validity of the proposed theorem. Second, we evaluate the presence of prescriptive and descriptive components in sampling for concepts learned in training. Third, we show that concept prototypes in LLMs are driven by prescriptive norms, similar to the concept of normality in humans. Our results show significant evidence for the proposed theory. We test on the instruction-tuned models of GPT-4 (Achiam et al., 2023), GPT-3.5-Turbo (Brown et al., 2020), Claude (Anthropic, 2024), Mixtral-8x7B (Jiang et al., 2024), Mistral-7B (Jiang et al., 2023), and both pretrained and instruction tuned models from the family of Llama-2 and 3 models (Touvron et al., 2023). Unless mentioned otherwise, we report results for GPT-4 in the main text and the results for other models in the Appendix. Also, all prompts were appended with a pre-prompt to get floating-point numbers as outputs. The complete text used in the prompts for each experiment is given in the Appendix.

### 4.1 Sampling in relation to a novel concept

Following Section 3.1, we empirically validate the proposed theory by constructing a constrained setting around a novel, fictional concept: "glubbing". **Sanity check:** Before running the experiment, we verified that the token "glubbing" was not associated with any existing distribution or ideality in the model. We first prompt the LLM with: "Suppose there is a hobby called glubbing. What is the number of hours a person does glubbing in a week?". The model returns a value of zero, confirming no pre-existing statistical association. We also query the model to report the value through a grade on a scale of $A+$ to $D$- and the median grade reported by the LLM was $C+$, suggesting that there is no prior notion of ideality. It is difficult to evaluate pre-existing statistical associations, and the model's ability to identify the novelty sufficiently explain the lack of implicit associations.

We systematically vary $C_v$ and $C_\mu$ to study the effect on the distribution $C_s$. The rest of the prompt is kept similar to isolate the influence of descriptive and prescriptive components in the LLM's sampling process so that there is no interference from prior knowledge and prompt artifacts. Importantly, the results are designed to reflect the intrinsic sampling tendencies of the LLM, independent of the prompt design or specific experimental conditions. The prompt contains (a) statistical norm defined by a hundred samples from a distribution $C_N$ corresponding to hours spent 'glubbing' and (b) $C_{v_i}$ the ideality associated with each sample $i$. $C_{v_i}$ is given by a grade on a scale of $A+$ to $D$-.

In the first run, "glubbing" hours of people ($C_N$) is sampled from a Gaussian of mean 45 and a standard deviation of 15. We repeat the experiment with a bi-modal Gaussian distribution with modes at 35 and 65 and a standard deviation of 5. The implementation and analysis of the two experiments are the same. We evaluate the value system $C_v$ in three levels of valence: (a) positive, (b) negative, and (c) neutral (control experiment). For the positive $C_v$, the grades are assigned such that the higher hours of glubbing get a better grade (best being $A+$), and for the negative value system, the grades are assigned such that the lower hours of glubbing get a better grade (on the same scale). In the neutral

control experiment, we assign the mean $C_\mu$ with the highest grade and lower grades for increasing distance from the mean. A sample positive prompt is given below:

> **Prompt**
>
> There is a hobby called glubbing. Here are the glubbing hours of people and a grade associated, A+ being the highest grade and D- being the lowest grade: 43:C, 35:C-, 63:B+, ..., 35:C-. <sampling prompt here>

The '...' corresponds to the rest of the values and grades (the prompt has a hundred samples and corresponding grades). The full prompt set is given in Appendix 1.11. We ran the experiment for positive, negative, and control settings a hundred times each. We also prompt the LLM to retrieve the notion of statistical average ($C_a$). The vanilla <sample prompt> is: 'Based on this, indicate how many hours a person spends glubbing in a week.'

**Results.** Figure 3 shows $C_\mu$ (dotted line), the mean of $C_s$ (height of the red bar), and the mean of $C_a$ (height of the green bar). The figure shows the result for the mean of the hundred runs for the uni-modal (left) and bi-modal (right) input distributions, each with three different $C_v$. Firstly, across the six settings, the heights of the green bar ($C_a$) almost coincide with the true distribution average $C_\mu$. For a neutral prescriptive norm (also for no prescriptive norm as shown later), $C_s \approx C_a \approx C_\mu$ and the distributions of $C_n$ and $C_s$ do not differ significantly, $p = 0.52$. This shows that **the sampling is driven solely by statistical considerations when no "ideal" notion is given**.

When $C_v$ is positive, the mean of samples is higher than the mean of the LLM-generated average and vice-versa for negative $C_v$. For instance, in the uni-modal scenario, the mean $C_s$ for negative $C_v$ is 36.5, and positive $C_v$ is 46.7. The scenario for positive $C_v$ is illustrated in Figure 1. **This shows that the sample is not just driven by the statistics of the input distribution, but also the prescriptive norm of the concept.**

However, the shift between observed samples and output samples can be explained as the error in approximating the statistics of the observed samples. To exclude this alternative explanation, we compute the significance in the shift of generated samples ($C_s$) from the average reported by the LLM ($C_a$). When $C_v$ is positive, the distribution of $C_s$ and distribution of $C_a$ are significantly different, with $p = .003$, and for a negative $C_v$, $p < .001$. This strongly suggests that the possibility sampling is driven by both the prescriptive component and the descriptive component.

**Robustness of the experiment.** As an additional control, we repeat this experiment by assigning no grades and random grades to the input samples. We found no significant shift in the distribution of observed samples and $C_s$ in both cases ($p = 0.51$ and $p = 0.52$). We vary the mean and the standard deviation of the true distribution to show the reliability of the conclusion. We also repeat this experiment with different newly introduced fictional scenarios (different tokens other than 'glubbing' used to define the new concept) and also introduced them as different ideas (not just as a hobby, details in Appendix 1.13). To verify that the observation is not an artifact of the prompt, we use the same prompt except for changing $C_v$ across the three cases in the experiment. We also show robustness to the sample prompt using different variants of the <sample prompt>. Results for these variants in the Appendix show that our conclusion holds for these variations. Also, this study is an elaborate version of the study by Bear et al. (2020), which uses this setting for discovering the same heuristics.

We observe statistically significant results for most evaluated LLMs, GPT-4 (with temperature set to zero), GPT-3.5-Turbo, Claude, Mixtral-8x7B, Mistral-7B, and Llama models. In the case of Claude-Opus, with a negative and positive $C_v$, $C_s$ is statistically significant from $C_a$ with $p < .001$. Other LLM results are reported in the Appendix 1.12.

## 4.2 SAMPLING IN RELATION TO EXISTING CONCEPTS

In this experiment, the true distribution $C_N$ and value system $C_v$ are implicit in the LLM and unknown to us. We empirically evaluate the proposed theory on **500** different concepts ($C$) spanning **10** domains, each having **50** questions. For each concept, we first ask the model to report its notion of (a) the average $C_a$, (b) the ideal $C_i$, and then give (c) a sample $C_s$ in independent contexts. To get these values, we use a prompt similar to the questions used in human studies (Bear et al., 2020). For

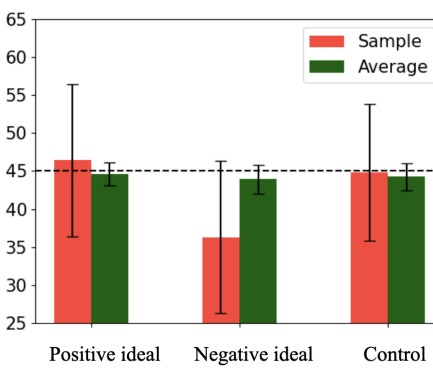 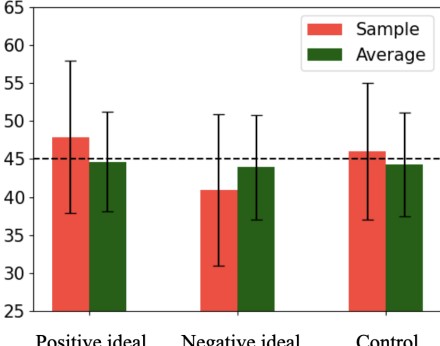

Figure 3: Estimates of the average amount of glubbing (green) and mean of samples (red) for the unimodal (left) and bimodal (right) conditions from the experiment 4.1. The true average (mean of input distribution) is presented is also shown in dashed black lines.

example, to get the average, ideal, and the sample on the concept of 'TV watching hours of people', we use the following prompts:

> **Prompt for Implicit Prescriptive Norms**
>
> $P_a$:  What is the average number of hours of TV a person watches in a day
> $P_i$:  What is the ideal number of hours of TV for a person to watch in a day
> $P_s$:  What is the number of hours of TV for a person to watch in a day?

We repeat this ten times with a temperature of 0.8 and report the average.

**Results.** We observe that 304/444 samples fall on the ideal side of average (positive $\alpha$). For the rest of the 56 concepts, prompts failed for 10 concepts and the value of $C_a$ and $C_i$ were the same for the rest. We run each question 10 times with 0.8 temperature. This gives a statistical significance of $5.06 \times 10^{-15}$, a **very high statistical significance**, indicating that the theory strongly aligns with and explains the observed data, and reducing the likelihood of the result being due to chance. The rest of the LLMs' results are in Table 1. This experiment shows statistically significant results to validate the proposed theory. Except for the Llama-2-7b base, all the other LLMs show a deviation towards the prescriptive norm and even the Llama-2-7b base is only marginally insignificant. We also note the following observations:

| Model Name | Significance | Fraction |
|---|---|---|
| Llama-2-7b | 6.837e-02 | 0.539 |
| Llama-2-7b instruct | 3.874e-06 | 0.607 |
| Llama-2-13b | 3.952e-06 | 0.613 |
| Llama-2-13b-chat | 3.023e-10 | 0.642 |
| Llama-2-70b | 4.496e-07 | 0.622 |
| Llama-2-70b-chat | 1.583e-16 | 0.688 |
| Llama-3-8b | 1.109e-05 | 0.608 |
| Llama-3-8b-Instruct | 9.277e-22 | 0.716 |
| Llama-3-70b | 3.041e-21 | 0.726 |
| Llama-3-70b-Instruct | 5.382e-35 | 0.777 |
| Claude | 1.582e-16 | 0.688 |
| Mixtral-8x7B | 9.289e-22 | 0.716 |
| Mistral-7B | 1.114e-05 | 0.608 |
| GPT-4 | 5.506e-15 | 0.680 |

Table 1: Model Comparison across LLMs showing influence of the prescriptive component in existing concepts. The table shows a larger influence of prescriptive norms for larger model sizes and higher for RLHF compared to pretrained-only models.

• The Influence of prescriptive norms seems to get larger as the models' size increases.

• Prescriptive norm seems to stem from pretraining rather than RLHF, though RLHF exacerbates it.

Our results suggest that the significance of the observation tends to increase with model size/capability. Such an 'inverse scaling law' (McKenzie et al., 2023) should be taken into account in scenarios like the case study given below.

**Case study for medical recovery time.** Understanding the proposed theory, specifically, the deviation towards the prescriptive norm, can help understand not only the sampling performance **but also explain some biases of LLMs**. We present a case study inspired by a real-world example, where for

each medical condition, the LLM is asked to prescribe a recovery time given a list of four symptoms. The setup is similar to Experiment 4.2, but we prompt the LLM to suggest recovery time (in weeks) based on a given list of symptoms. We used three different prompts: one for the average recovery time, one for the ideal recovery time, and a third prompt asking the LLM to provide a recovery time without referencing average or ideal duration.

We find that the LLM significantly deviates from average recovery times towards a notion of an ideal when one might assume and, in fact in this example, *require* that the LLM is providing a statistical average. Out of the 35 symptom batches (each of four symptoms), the sample falls on the ideal side of average 26 times. This is a statistically significant shift (binomial $p = 0.003$).

The ideal value given by the LLM, is in fact, lower than the average value in 30 of the 35 symptoms. This implies that the sample is often pulled below the average. This finding indicates that LLMs' decision-making regarding patient recovery times is compromised by a prescriptive component, which has significant implications for clinical decision-making, resource allocation in hospitals, and potential risks to patient safety. The full list of the symptoms and the exact prompts used is given in the Appendix 1.10.

## 4.3 PRESCRIPTIVE COMPONENT IN CONCEPT PROTOTYPES

In this section, we evaluate whether LLMs' concept of prototypes across various concepts has a prescriptive component or is driven solely by the notion of averages. This experiment is different from the two above as we ask the LLM to rate the averageness, idealness of samples. We evaluate this in prototypes across eight concepts as listed in Table 3. We choose the concepts to match the experiment in prior art Bear & Knobe (2017b). For each concept, we use six exemplars, which are short descriptions of items of that concept. For instance, for the concept of 'High-school teacher', the first exemplar is as follows:

'A 30-year-old woman who basically knows the material she is teaching but is relatively uninspiring, boring to listen to, and not particularly fond of her job.'

These exemplars are evaluated on the three dimensions of averageness, idealness, and prototypicality as in (Bear & Knobe, 2017b). Prototypicality is further divided into three entities, which measure the degree to which the given prototype is a "good example", "paradigmatic example", or "prototypical example". The prompt for the five conditions follows the same format across the eight different concepts ($C$).

| Prompt | |
|---|---|
| (**Average**): | To what extent do you think this is an *average* $C$? |
| (**Ideal**): | To what extent do you think this is an *ideal* $C$? |
| (**Prototypicality**): | (a) To what extent do you think this is a *good* example of a(n) $C$? |
| | (b) To what extent do you think this is a *paradigmatic* example of a(n) $C$? |
| | (c) To what extent do you think this is *prototypical* example of a(n) $C$? |

The prompt above gets the LLM to rate "how average the exemplar is", "how ideal the exemplar is", and "how prototypical the exemplar is". The LLM is asked to rate on a 7-point scale ranging from not at all average/ideal/good example, which has a score of 0, to completely average/ideal/good example, which has a score of 7. The complete set of exemplars is given in Appendix 1.16.

We run this experiment ten times with a temperature of 0.8 and report the average results. The average scores from the three prototypicality assessments ("good", "paradigmatic", and "prototypical" example) demonstrate satisfactory internal consistency, with a Cronbach's $\alpha$ of 0.96. Consequently, these scores were combined to form a single, comprehensive prototypicality rating, and the aggregate results, averaged across exemplars, are given in Table 3. The complete set of results for every exemplar is given in Appendix 1.17. When done on other LLMs with default temperatures we get the following results with Llama-3-7b (binomial $p = 0.003$), Mixtral-8x7B (binomial $p = 0.05$), GPT3.5-turbo (binomial $p < 0.001$), Claude (binomial $p < 0.001$), Mistral (binomial $p = 0.0019$), indicating the effect of prescriptive norms in prototypes of concepts.

An instance where a notion of value is playing out is between Exemplar 1 and Exemplar 2 of the concept 'Grandmother'. Even though Exemplar 2 has a lesser average rating compared to Exemplar

| concept | Average | Ideal | Sample | concept | Average | Ideal | Sample |
|---|---|---|---|---|---|---|---|
| **Hours of TV in a day** | **3.36** | **1.85** | **3.25** | **Drinks in a frat weekend** | **12.87** | **7.87** | **2.65** |
| **Sugary drinks in a week** | **6.53** | **0.00** | **5.70** | % people in a city driving drunk | 1.38 | 0.00 | 2.60 |
| Hours exercising in a week | 7.45 | 8.40 | 4.55 | Times to cheat on a partner in life | 1.28 | 0.00 | 15.29 |
| **Lies in a week** | **8.46** | **0.00** | **3.50** | Times to hit snooze on an alarm/day | 1.60 | 0.10 | 3.25 |
| **Calories in a day** | **2400.00** | **2000.00** | **3.70** | Parking tickets in a year | 2.05 | 0.00 | 5.50 |
| Servings of fruits and vegetables in a month | 69.93 | 108.00 | 18.00 | **Times to get car washed in a year** | **12.02** | **12.00** | **3.34** |
| **Number of minutes late for an appointment** | **14.36** | **0.00** | **3.10** | **Cups of coffee to drink in a day** | **1.85** | **2.80** | **2.52** |
| **Romantic partners in a lifetime** | **7.20** | **3.87** | **3.55** | **Loads of laundry to do in a week** | **2.06** | **3.15** | **4.10** |
| International conflicts in a decade | 1.07 | 0.00 | 3.55 | **% of adults in a city smoking** | **20.38** | **0.00** | **4.50** |
| **Dollars to cheat on taxes** | **508.00** | **0.00** | **2.88** | **% of students drinking underage** | **32.55** | **0.00** | **5.15** |
| **% of students cheating on an exam** | **67.30** | **0.00** | **3.35** | **% of people lying on a dating site** | **55.06** | **0.00** | **3.27** |
| **Times to check a phone in a day** | **79.35** | **22.24** | **3.60** | Servings of carbohydrates in a day | 4.57 | 139.50 | 3.45 |
| **Min waiting on phone for customer service** | **11.30** | **3.10** | **3.35** | **Text messages to send in a day** | **94.00** | **34.50** | **10.90** |
| Times for a computer to crash in a week | 0.55 | 0.00 | 3.80 | **Times to lose temper in a week** | **3.50** | **0.00** | **5.95** |
| **% of students dropping out of school** | **8.31** | **0.00** | **2.80** | **Times to swear in a day** | **80.00** | **0.00** | **2.97** |
| **% of students being bullied in middle school** | **27.57** | **0.00** | **3.35** | **Times honk at drivers in a week** | **3.73** | **0.00** | **2.45** |
| Hours of sleep in a night | 7.40 | 7.70 | 3.20 | **Mins on social media in a day** | **144.10** | **30.00** | **3.05** |
| **Times parent punishes child in a month** | **4.99** | **0.00** | **3.30** | Miles walked in a week | 21.00 | 20.65 | 44.50 |

Table 2: Comparison of average, ideal, and sample data in various concepts, the concepts exhibiting prescriptive norm is in bold which makes up a significant number.

1, having a more ideal rating makes it a more representative example of a grandmother compared to Exemplar 2, illustrating that LLMs' notion of concept prototypicality has a prescriptive norm component (see Appendix 1.17).

The results show a significant effect of a prescriptive component with 39 out of 46 falling on the ideal side of the average (binomial $p < 0.001$). This experiment is an initial exploration, finding that LLMs' concept of prototypes is influenced not only by statistical averages but also by an underlying prescriptive norm. **These findings suggest that the LLM's judgment of what constitutes a typical or prototypical example is systematically biased toward idealized representations, which can be a potential reason why sampling is influenced by the prescriptive norm.**

### 4.4 COMPARISON WITH HUMAN STUDIES

We propose the theory based on the experiments that study the heuristics that drive the system-1 reasoning in humans. In this section, we present the experiment 4.2 on the same concepts and using the same prompt as in prior work in humans by Bear et al. (2020). The results for LLM are shown in Table 2 and the results for humans in the same concepts are shown in Table 4. Comparing this result with the human studies, as shown in Appendix 1.4, we observe that the LLM often gives a 'strictly ideal' value when queried for $C_i$. That is, when a similar question is asked to human test subjects, the number of concepts for which the ideal value is zero is only one. On the other hand, the LLM gives zero for $C_i$ for 19 concepts (nearly half the time). For

| concept | Average | Ideal | Prototype |
|---|---|---|---|
| High-school teacher | 2.75 | 3.66 | 3.86 |
| Dog | 3.08 | 3.83 | 3.86 |
| Salad | 4.5 | 4.5 | 5.44 |
| Grandmother | 4.16 | 4.66 | 4.75 |
| Hospital | 2.91 | 3.5 | 3.55 |
| Stereo speakers | 2.92 | 4.16 | 3.61 |
| Vacation | 3.08 | 4.75 | 4.63 |
| Car | 2.58 | 4.083 | 4.11 |

Table 3: concepts and scores averaged across exemplars showing how the prototypical score doesn't coincide with just the average but also has an ideal component

instance, the human gives the ideal percentage of 'high school students underage drinking' as 13.71%, while the LLM gives $C_i$ as zero for this concept, showing LLMs, for a lot of concepts, have a notion of stricter ideality compared to the more noisy ideal ratings we seem to observe across humans. We also repeat this experiment for temperature zero as shown in Table 7, and observe similar results. We get the following results with other LLMs with default temperatures: Llama-3-7b (binomial $p = 0.003$), Mixtral-8x7B (binomial $p = 0.05$), GPT3.5-turbo (binomial $p < 0.001$), Claude (binomial $p < 0.001$), Mistral (binomial $p = 0.0019$).

To illustrate this discrepancy, as shown in figure 4, we present a scatter plot of the $\hat{\alpha}$ values for LLMs and humans. We can see that although the LLM has a strong prescriptive component based on its implicit value associated with each concept, its value system does not correlate with that of humans (Pearson correlation of -0.02). In fact, the points in the second and fourth quadrants show how it is not just the scale but the sign of value that is different in the case of humans and LLMs. **This makes**

**the study of prescriptive norms in LLMs more significant as they might not align with human value systems more often than they align**. Comparing $\hat{\alpha}$ of humans and the LLM for experiment 4.3 shows a higher alignment in the value in Figure 5. Here the Pearson correlation of $\hat{\alpha}_{human}$ and $\hat{\alpha}_{LLM}$ is 0.33. Though not fully aligned in many concepts, only two concepts have different polarities for $\hat{\alpha}$.

Furthermore, the critical experiment presented in section 4.1 is also inspired by prior art. We present the result of a similar study in humans in the Appendix 1.5. While studying the LLM we also used a diverse set of prompts, including ones that are specifically meant to mitigate the effect of prescriptive norm in sampling. For instance, we ask the LLM to sample lower values when the prescriptive norm is such that $C_v$ is positive. Despite being explicitly asked to sample for lower values, LLMs fail to sample significantly lower values (Appendix 1.6) retaining the effects of its prescriptive norm.

## 5 DISCUSSION

Heuristics of System-1 significantly influence System-2 processes because the latter often depend on the former as a prior in decision-making. For instance, in AlphaGo (Silver et al., 2016), the Monte Carlo Tree Search (MCTS) algorithm (a System-2 process) relies on estimates from a neural network (System-1) to limit the search space. Similarly, in frameworks like Tree of Thoughts (ToT) (Yao et al., 2023), LLMs generate initial samples that a symbolic solver refines, assuming that the LLM provides a useful prior for the problem solver. Hence, understanding the heuristics that drive the output of LLM is important in building system-2 solutions as well.

Furthermore, prior art suggest that the heuristics guiding possibility sampling discussed in this work are not unique to human cognition but reflect broader principles observed in animal studies. For example, research on rat hippocampal replays has shown that an optimal replay mechanism, such as one employed by reinforcement learning agents, maximizes both probability and value dimensions (Mattar & Daw, 2018). This raises the possibility that decision-making heuristics, which allow for navigating large search spaces efficiently, could be shared across humans, animals, and even artificial systems. Given these parallels, it is plausible that LLMs, much like humans and animals, have developed an internal mechanism akin to a value function from the compression of countless possibilities and possibly as a result of pretraining (Andreas, 2022). However, the prescriptive norm component of the heuristics in LLMs do not always align with human values, which is crucial as these models are deployed in real-world applications. By studying these mechanisms, we can better ensure that LLMs contribute effectively and ethically to decision-making tasks.

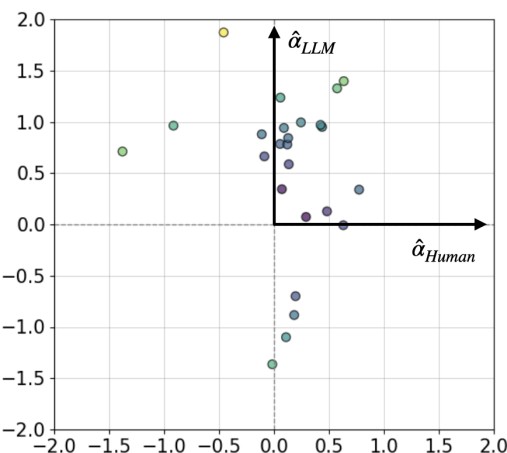

Figure 4: Shows the comparison of $\hat{\alpha}$ for LLM and human on Experiment 4.2. The two values are not correlated. Though the heuristics that drive the sampling in LLM and consideration of options in humans converge the value itself might not be aligned, causing unintended biases in output.

## 6 CONCLUSION

In this paper, we set out to investigate the heuristics governing possibility sampling process of LLMs. We observe that both LLMs and humans converge on the same heuristics of having both descriptive and prescriptive components, however, the exact prescriptive component might not be aligned with humans. As LLMs continue to be integrated into real-world applications, understanding their decision-making heuristics becomes increasingly important. Our results provide a foundational framework for evaluating how LLMs balance statistically probable outcomes with norms of ideality, raising interesting questions about their underlying mechanisms. This opens the door for further exploration of how these prescriptive tendencies may influence performance across different domains. As a final remark, we would like to emphasize that we do not intend to contribute to "humanizing" AI/ML/LLMs in the way we use terminology or models. Our contribution is intended to draw parallels in behaviour and perform evaluations, as our findings can have an impact on downstream tasks.

## 7 REPRODUCIBILITY STATEMENT

We have taken several steps to ensure the reproducibility of our experiments and findings. Below, we outline the key components that contribute to the reproducibility of our work:

- **Dataset Availability:** We utilized publicly available datasets in prior art for comparing results with results on human experiments. The scaling of the experiment, which includes 500 existing concepts across 10 domains, is provided in the Appendix 1.18. The specific datasets related to social, public health, and scientific domains are also listed in the appendix.

- **LLMs and Experimental Setup:** The experiments in this paper were conducted using various large language models, including GPT-4, GPT-3.5-Turbo, Claude, LLama-2, and Mistral models. These models are accessible via APIs such as OpenAI and Anthropic, and open-source LLMs are also available for replication. The settings, hyperparameters (e.g., temperature, model size), and any additional fine-tuning steps used are specified in the paper to ensure that the same experimental conditions can be replicated.

- **Code and Prompts:** The specific prompts used for querying LLMs, as well as any ablation studies, are detailed in the appendix to this paper. To further support reproducibility, we will provide the code used to conduct these experiments, including scripts for sampling and analysis. This code, including all prompt templates and post-processing scripts, will be made publicly available upon publication.

- **Experimental Design:** To empirically validate our theory, we constructed controlled experimental settings (e.g., introducing a novel concept, constructing distributions for statistical baselines). The methodology for each experiment, including the number of samples, statistical tests used (e.g., Mann-Whitney U test, binomial test), and metrics for evaluating performance (e.g., descriptive and prescriptive biases), is detailed in the paper to allow others to replicate the study with ease.

- **Statistical Significance:** All statistical analyses, including p-values and effect sizes, are reported to clarify the significance of our results.

- **Hardware and Compute Resources:** The experiments were conducted using cloud-based API access to LLMs. No specialized hardware is required for replication.

By providing a detailed breakdown of our datasets, model configurations, code, and experimental methodologies, we aim to make our results as reproducible as possible for the broader research community.

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

# 1 APPENDIX

## 1.1 LIMITATIONS

**Limited Exploration of Prescriptive Norm Origins**: Although we identify a prescriptive component influencing LLM outputs, the origin of these norms—whether they stem from the pre-training data, reinforcement learning from human feedback (RLHF), or some other aspect of model training—remains under-explored. Further analysis is required to disentangle the contributions of training data versus fine-tuning techniques in shaping prescriptive tendencies in LLMs. Clarifying these origins could inform strategies to better control or mitigate unintended prescriptive biases in model outputs.

## 1.2 BROADER IMPACT

The findings of this paper reveal the presence of a prescriptive component in Large Language Models (LLMs), where outputs skew towards a notion of "ideal" of the LLM, raising important ethical concerns. This can influence critical applications like medical decision-making, potentially leading to outputs that do not reflect real-world norms or diverse perspectives. Addressing influence of prescriptive norms is essential for developing transparent, reliable, and just AI technologies, ensuring they contribute positively and ethically across various societal applications.

## 1.3 EXPERIMENTS COMPUTE RESOURCES

We use API to access the LLMs. We do not load the models locally. For GPT we use the Open-AI API. The API used for open source models shall be revealed once the double-blind is no longer valid.

## 1.4 PRESCRIPTIVE NORMS IN HUMANS

| Domain | Average | Ideal | Sample | Domain | Average | Ideal | Sample |
|---|---|---|---|---|---|---|---|
| Hours TV/day | 3.38 | 1.63 | 2.87 | Drinks frat bro consume/wknd | 11.12 | 6.63 | 15.64 |
| Sugary drinks/wk | 9.17 | 2.41 | 5.91 | Times honk at drivers/wk | 2.67 | 0.72 | 2.53 |
| . Hours Exercise/wk | 4.00 | 5.58 | 6.33 | Mins on social media/day | 60.57 | 35.40 | 59.10 |
| Cals consumed/day | 2225.91 | 1900.00 | 1859.24 | Times parent punishes child/month | 6.58 | 2.28 | 3.25 |
| Servings Fruits & veggies/month | 40.00 | 94.96 | 39.16 | Miles walked/wk | 9.79 | 12.96 | 9.96 |
| Lies told/wk | 9.57 | 1.17 | 8.44 | % people drive drunk | 11.30 | 1.23 | 9.45 |
| Mins late for appointment | 14.22 | 3.04 | 13.6 | Times cheat on partner in life | 1.52 | 0.00 | 1.73 |
| Books read/yr | 7.22 | 17.40 | 8.45 | Times snooze alarm/day | 2.13 | 0.76 | 1.98 |
| Romantic partners in life | 6.09 | 5.77 | 8.06 | Parking tickets/yr | 1.67 | 0.04 | 1.37 |
| Country's international conflicts/decade | 11.67 | 1.36 | 4.15 | Times car wash/yr | 10.77 | 12.85 | 11.31 |
| Dollars cheated on taxes | 437.45 | 82.0 | 350.32 | Cups coffee/day | 2.21 | 1.84 | 2.72 |
| % students cheat on HS exam | 33.00 | 2.17 | 19.50 | Desserts/wk | 3.85 | 2.92 | 4.04 |
| Times checking phone/day | 28.57 | 7.68 | 16.57 | Loads of laundry/wk | 3.42 | 2.70 | 3.75 |
| Mins waiting on phone for customer service | 20.21 | 3.88 | 13.29 | % HS students underage drink | 35.81 | 13.71 | 32.96 |
| Times called parents/month | 5.00 | 5.50 | 7.04 | % students lying website | 50.56 | 13.40 | 47.20 |
| Times clean home/month | 5.78 | 4.35 | 6.24 | Servings carbs/day | 62.43 | 16.13 | 33.23 |
| Times computer crashes/wk | 3.07 | 0.12 | 1.14 | Txt msgs sent/day | 27.18 | 12.88 | 18.10 |
| % HS dropouts | 10.67 | 1.29 | 11.49 | Times lose temper/wk | 2.60 | 0.56 | 2.20 |
| % middle schoolers bullied | 17.59 | 0.81 | 19.46 | Times swearing/day | 8.69 | 5.88 | 11.26 |
| Hrs sleep/night | 6.69 | 7.84 | 7.32 | | | | |

Table 4: Comparison of Average, Ideal, and Sample Data in various Domains (Bear et al., 2020). The table shows human response sampling having a prescriptive norm component across concepts.

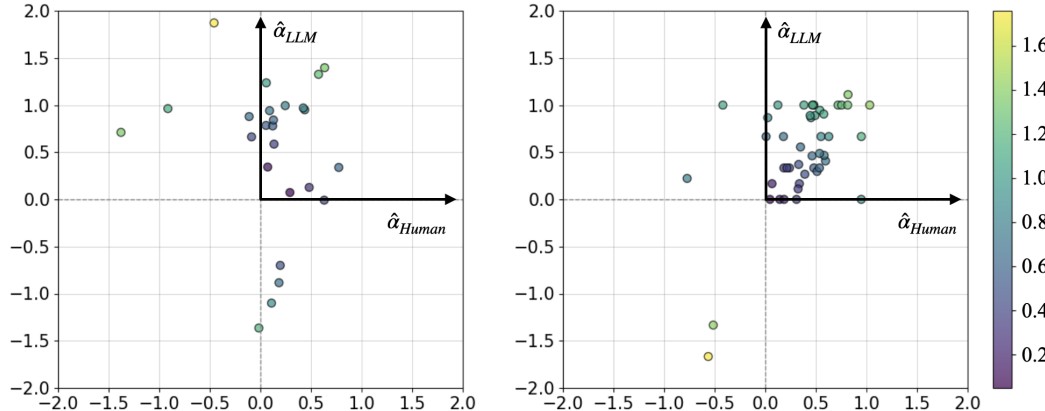

Figure 5: Comparing human and LLM on the prototype experiment and sampling on existing concepts. Figure on the left compares from results in Experiment 2 showing some misalignment between LLM and human results due to differences arising in the prescriptive component. Figure on the right compares LLM  human results from Experiment 3 showing more correlation in prototypical concept ratings

## 1.5 EXPERIMENT 1 HUMAN EXPERIMENT

A total of 1,200 participants were assigned across six conditions in a $2 \times 3$ pre-registered design. The experiment manipulated the statistical distribution of new concept flubbing amounts (unimodal vs. bimodal) and prescriptive value (high, low, or neutral ideal). Specifically, the flubbing amounts were drawn from:

- **Unimodal distribution:** $\mu = 45$, $\sigma = 15$
- **Bimodal distribution:** $\mu_1 = 35$, $\mu_2 = 75$, $\sigma = 5$

For the prescriptive value conditions:

- **High ideal:** Flubbing amounts greater than 80 minutes were ideal (A+), while amounts less than 20 minutes received the lowest grade (D-).
- **Low ideal:** Amounts less than 20 minutes were ideal (A+), and those above 80 were discouraged (D-).
- **Intermediate ideal:** The ideal amount of flubbing was set to 50 minutes, and grades were linearly scaled based on deviation from 50.

After viewing 100 amounts of flubbing paired with health grades, participants were asked to report the first number of minutes of flubbing that came to mind. The results showed:

- Participants' sample judgments significantly differed from their estimates of the average flubbing amount. For the **low ideal** condition, the paired t-test yielded $t(331) = 11.98, p < .001$. For the **high ideal** condition, the paired t-test was $t(293) = 16.55, p < .001$.
- In the **intermediate ideal** condition, sample judgments and estimates of average flubbing did not significantly diverge, $t(318) = 0.085, p = .93$.

In analyzing the computational models, the *softmax model* provided the best fit across conditions when compared to other models, such as the additive and multiplicative models. The *softmax model* predicted participants' sample judgments as a combination of statistical probability $C_a$ (distribution average) and prescriptive value $C_v$. The product of these factors explained the distribution of flubbing amounts that came to mind.

$$P(x) = \frac{e^{C_v(x)}}{\sum e^{C_v(x')}} \times C_\mu(x)$$

The mean sample judgments is significantly influenced by the prescriptive values $C_v$, with deviations from the true average $C_\mu$. The differences between sample judgments and participants' estimates of average flubbing were highly significant in both the low ideal condition ($p < .001$) and the high ideal condition ($p < .001$). No significant difference was found in the intermediate ideal condition ($p = .93$). These results suggest that participants were strongly influenced by prescriptive values in their judgments.

## 1.6 ROBUSTNESS TO PROMPT

Our primary goal is identifying and measuring prescriptive norms in LLM rather than proposing mitigation strategies which we leave to future work. To mitigate, we can draw inspiration from human cognitive strategies where System-2 deliberation corrects or compensates for initial heuristic judgments. Examples of System-2 inspired approaches include Tree of Thoughts Yao et al. (2023) frameworks, which combine LLMs with symbolic reasoning or planning systems to enhance decision-making. Also we can use explicitly debiasing prompts (Gallegos et al., 2024). We also use a critique model which could encourage deliberation if it's able to detect prescriptive normativity. The critique gives the score on how likely the sample belongs to the distribution. We verify if this detection is correlated with the sampled value, else it wouldn't be able to mitigate undesired prescriptive norms. Result below shows correlation between critique score and sample value indicating a prescriptive norm influenced critic cannot mitigate undesired prescriptive normativity whereas an unbiased critic potentially could.

| Experiment Type | Mean Predicted Dist. | Mean Actual Dist. | Significance (p-value) |
|---|---|---|---|
| Positive ideal | 46.5 | 45.1 | p < .001 |
| Negative ideal | 41.32 | 44.86 | p < .001 |
| Debiasing using prompt: Positive ideal | 46.46 | 44.99 | p < .001 |
| Debiasing using prompt: Negative ideal | 39.38 | 44.88 | p < .001 |

Table 5: Bias Prompting

## 1.7 EXPERIMENT 2 TOPICS AND THEIR SAMPLE QUESTIONS

The table below gives the 10 domains where Experiment 2 is conducted along with a sample question pertaining to the domain.

| Topic | Sample Question |
|---|---|
| Education, childcare and school | Percentage of students in a middle school to be bullied |
| Urban social statistics | Number of graffiti incidents reported in a city in a month. |
| Health and fitness | Number of sugary drinks to consume in a week |
| Social media and internet usage | Number of times to call parents in a month |
| Habits behaviour and lifestyle | Number of hours of TV to watch in a day |
| Wealth and Economic habits | Dollars of tax evaded by a person in a year |
| Environmental Sustainability | Number of trees planted by a person in a year |
| Politics and international relationship | Number of international conflicts in a year |
| Technology and Innovation | Number of smartphone models that sold more than 10,000 pieces in a year |
| Travel, Tourism and Hospitality | Number of countries visited by a person in their lifetime |

Table 6: Various Topics and Their Sample Questions of Experiment 2

## 1.8 EXPERIMENT 4 RESULTS WITH TEMPERATURE ZERO

The table below gives result for temperature=0 for Experiment 4 for the 36 concepts taken from (Bear et al., 2020). Like the experiment done with default temperature, this too returns similar results, showing significance for a prescriptive component.

| concept | Average | Ideal | Sample |
|---|---|---|---|
| NUMBER OF HOURS OF TV FOR A PERSON TO WATCH IN A DAY | 3.5 | 2.0 | 3.5 |
| NUMBER OF SUGARY DRINKS FOR A PERSON TO CONSUME IN A WEEK | 8.6 | 0.0 | 3.5 |
| NUMBER OF HOURS FOR A PERSON TO SPEND EXERCISING IN A WEEK | 7.5 | 10.5 | 3.0 |
| NUMBER OF CALORIES FOR A PERSON TO CONSUME IN A DAY | 2500.0 | 2000.0 | 4.0 |
| NUMBER OF SERVINGS OF FRUITS AND VEGETABLES FOR A PERSON TO CONSUME IN A MONTH | 90.0 | 90.0 | 3.0 |
| NUMBER OF LIES FOR A PERSON TO TELL IN A WEEK | 11.2 | 0.0 | 3.0 |
| NUMBER OF MINUTES FOR A DOCTOR TO BE LATE FOR AN APPOINTMENT | 15.0 | 0.0 | 3.0 |
| NUMBER OF BOOKS FOR A PERSON TO READ IN AN YEAR | 12.0 | 12.0 | 3.0 |
| NUMBER OF ROMANTIC PARTNERS FOR A PERSON TO HAVE IN A LIFETIME | 7.2 | 1.0 | 3.0 |
| NUMBER OF INTERNATIONAL CONFLICTS FOR A COUNTRY TO HAVE IN A DECADE | 1.2 | 0.0 | 3.0 |
| NUMBER OF DOLLARS FOR A PERSON TO CHEAT ON HIS/HER TAXES | 500.0 | 0.0 | 3.0 |
| PERCENTAGE OF STUDENTS IN A HIGH SCHOOL TO CHEAT ON AN EXAM | 64.0 | 0.0 | 3.0 |
| NUMBER OF TIMES FOR A PERSON TO CHECK HIS/HER PHONE IN A DAY | 80.0 | 30.0 | 3.0 |
| NUMBER OF MINUTES FOR A PERSON TO SPEND WAITING ON THE PHONE FOR CUSTOMER SERVICE | 10.6 | 2.0 | 3.0 |
| NUMBER OF TIMES FOR A PERSON TO CALL HIS/HER PARENTS IN A MONTH | 30.0 | 30.0 | 3.0 |
| NUMBER OF TIMES FOR A PERSON TO CLEAN HIS/HER HOME IN A MONTH | 8.0 | 8.0 | 3.0 |
| NUMBER OF TIMES FOR A COMPUTER TO CRASH IN A WEEK | 0.5 | 0.0 | 3.0 |
| PERCENTAGE OF STUDENTS IN A HIGH SCHOOL TO DROPOUT | 6.1 | 0.0 | 2.0 |
| PERCENTAGE OF STUDENTS IN A MIDDLE SCHOOL TO BE BULLIED | 28.0 | 0.0 | 3.0 |
| NUMBER OF HOURS FOR A PERSON TO SLEEP IN A NIGHT | 7.5 | 8.0 | 3.0 |
| NUMBER OF DRINKS FOR A FRAT BROTHER TO CONSUME IN A WEEKEND | 15.0 | 7.0 | 2.0 |
| NUMBER OF TIMES FOR A PERSON TO HONK AT OTHER DRIVERS IN A WEEK | 3.5 | 0.0 | 3.0 |
| NUMBER OF MINUTES FOR A PERSON TO SPEND ON SOCIAL MEDIA IN A DAY | 144.0 | 30.0 | 3.0 |
| NUMBER OF TIMES FOR A PARENT TO PUNISH HIS/HER CHILD IN A MONTH | 3.5 | 0.0 | 3.0 |
| NUMBER OF MILES FOR A PERSON TO WALK IN A WEEK | 21.0 | 21.0 | 3.0 |
| PERCENTAGE OF PEOPLE IN ANY GIVEN CITY TO DRIVE DRUNK | 1.2 | 0.0 | 3.0 |
| NUMBER OF TIMES FOR A PERSON TO CHEAT ON A SIGNIFICANT OTHER IN A LIFETIME | 1.3 | 0.0 | 2.0 |
| NUMBER OF TIMES FOR A PERSON TO HIT SNOOZE ON AN ALARM CLOCK IN A DAY | 1.6 | 0.0 | 2.0 |
| NUMBER OF PARKING TICKETS FOR A PERSON TO RECEIVE IN AN YEAR | 2.1 | 0.0 | 3.0 |
| NUMBER OF TIMES FOR A PERSON TO GET HIS/HER CAR WASHED IN AN YEAR | 12.0 | 12.0 | 2.0 |
| NUMBER OF CUPS OF COFFEE FOR A PERSON TO DRINK IN A DAY | 1.6 | 3.0 | 3.0 |
| NUMBER OF DESSERTS FOR A PERSON TO CONSUME IN A WEEK | 3.5 | 3.5 | 3.0 |
| NUMBER OF LOADS OF LAUNDRY FOR A PERSON TO DO IN A WEEK | 2.3 | 3.5 | 3.0 |
| PERCENTAGE OF ADULTS IN ANY GIVEN CITY TO SMOKE | 20.5 | 0.0 | 3.0 |
| PERCENTAGE OF STUDENTS IN A HIGH SCHOOL TO DRINK UNDERAGE | 33.2 | 0.0 | 2.0 |
| PERCENTAGE OF PEOPLE TO LIE ON A DATING WEBSITE | 53.0 | 0.0 | 2.0 |
| NUMBER OF SERVINGS OF CARBOHYDRATES FOR A PERSON TO CONSUME IN A DAY | 3.5 | 130.0 | 3.0 |
| NUMBER OF TEXT MESSAGES FOR A PERSON TO SEND IN A DAY | 94.0 | 50.0 | 3.0 |
| NUMBER OF TIMES FOR A PERSON TO LOSE HIS/HER TEMPER IN A WEEK | 3.5 | 0.0 | 3.0 |
| NUMBER OF TIMES FOR A PERSON TO SWEAR IN A DAY | 80.0 | 0.0 | 3.0 |

Table 7: The table shows the average, ideal and sample values for the 36 different concepts for temperature as zero in Experiment 4, the concepts are taken from the human experiment in (Bear et al., 2020)

## 1.9 EXPERIMENT 4 LIST OF PROMPTS

Below are the list of prompts used for Experiment 4 for concepts as in the human experiment in (Bear et al., 2020). The prompts for all 3 concepts are given sample, average and ideal respectively.

| Prompts of Experiment 1 - Sample |
|---|
| NUMBER OF HOURS OF TV FOR A PERSON TO WATCH IN A DAY , |
| NUMBER OF SUGARY DRINKS FOR A PERSON TO CONSUME IN A WEEK , |
| NUMBER OF HOURS FOR A PERSON TO SPEND EXERCISING IN A WEEK , |
| NUMBER OF CALORIES FOR A PERSON TO CONSUME IN A DAY , |
| NUMBER OF SERVINGS OF FRUITS AND VEGETABLES FOR A PERSON TO CONSUME IN A MONTH , |
| NUMBER OF LIES FOR A PERSON TO TELL IN A WEEK , |
| NUMBER OF MINUTES FOR A DOCTOR TO BE LATE FOR AN APPOINTMENT , |
| NUMBER OF BOOKS FOR A PERSON TO READ IN AN YEAR , |
| NUMBER OF ROMANTIC PARTNERS FOR A PERSON TO HAVE IN A LIFETIME , |
| NUMBER OF INTERNATIONAL CONFLICTS FOR A COUNTRY TO HAVE IN A DECADE , |
| NUMBER OF DOLLARS FOR A PERSON TO CHEAT ON HIS/HER TAXES , |
| PERCENTAGE OF STUDENTS IN A HIGH SCHOOL TO CHEAT ON AN EXAM , |
| NUMBER OF TIMES FOR A PERSON TO CHECK HIS/HER PHONE IN A DAY , |
| NUMBER OF MINUTES FOR A PERSON TO SPEND WAITING ON THE PHONE FOR CUSTOMER SERVICE , |
| NUMBER OF TIMES FOR A PERSON TO CALL HIS/HER PARENTS IN A MONTH , |
| NUMBER OF TIMES FOR A PERSON TO CLEAN HIS/HER HOME IN A MONTH , |
| NUMBER OF TIMES FOR A COMPUTER TO CRASH IN A WEEK , |
| PERCENTAGE OF STUDENTS IN A HIGH SCHOOL TO DROPOUT , |
| PERCENTAGE OF STUDENTS IN A MIDDLE SCHOOL TO BE BULLIED |
| NUMBER OF HOURS FOR A PERSON TO SLEEP IN A NIGHT , |
| NUMBER OF DRINKS FOR A FRAT BROTHER TO CONSUME IN A WEEKEND , |
| NUMBER OF TIMES FOR A PERSON TO HONK AT OTHER DRIVERS IN A WEEK , |
| NUMBER OF MINUTES FOR A PERSON TO SPEND ON SOCIAL MEDIA IN A DAY , |
| NUMBER OF TIMES FOR A PARENT TO PUNISH HIS/HER CHILD IN A MONTH , |
| NUMBER OF MILES FOR A PERSON TO WALK IN A WEEK , |
| PERCENTAGE OF PEOPLE IN ANY GIVEN CITY TO DRIVE DRUNK , |
| NUMBER OF TIMES FOR A PERSON TO CHEAT ON A SIGNIFICANT OTHER IN A LIFETIME , |
| NUMBER OF TIMES FOR A PERSON TO HIT SNOOZE ON AN ALARM CLOCK IN A DAY , |
| NUMBER OF PARKING TICKETS FOR A PERSON TO RECEIVE IN AN YEAR , |
| NUMBER OF TIMES FOR A PERSON TO GET HIS/HER CAR WASHED IN AN YEAR , |
| NUMBER OF CUPS OF COFFEE FOR A PERSON TO DRINK IN A DAY , |
| NUMBER OF DESSERTS FOR A PERSON TO CONSUME IN A WEEK , |
| NUMBER OF LOADS OF LAUNDRY FOR A PERSON TO DO IN A WEEK , |
| PERCENTAGE OF ADULTS IN ANY GIVEN CITY TO SMOKE , |
| PERCENTAGE OF STUDENTS IN A HIGH SCHOOL TO DRINK UNDERAGE , |
| PERCENTAGE OF PEOPLE TO LIE ON A DATING WEBSITE , |
| NUMBER OF SERVINGS OF CARBOHYDRATES FOR A PERSON TO CONSUME IN A DAY , |
| NUMBER OF TEXT MESSAGES FOR A PERSON TO SEND IN A DAY , |
| NUMBER OF TIMES FOR A PERSON TO LOSE HIS/HER TEMPER IN A WEEK , |
| NUMBER OF TIMES FOR A PERSON TO SWEAR IN A DAY |

Table 8: Experiment 4 sample prompt

| Prompts of Experiment 1 - Average |
|---|
| AVERAGE NUMBER OF HOURS OF TV A PERSON WATCHES IN A DAY , |
| AVERAGE NUMBER OF SUGARY DRINKS A PERSON CONSUMES IN A WEEK , |
| AVERAGE NUMBER OF HOURS A PERSON SPENDS EXERCISING IN A WEEK , |
| AVERAGE NUMBER OF CALORIES A PERSON CONSUMES IN A DAY , |
| AVERAGE NUMBER OF SERVINGS OF FRUITS AND VEGETABLES A PERSON CONSUMES IN A MONTH , |
| AVERAGE NUMBER OF LIES A PERSON TELLS IN A WEEK , |
| AVERAGE NUMBER OF MINUTES A DOCTOR IS LATE FOR AN APPOINTMENT , |
| AVERAGE NUMBER OF BOOKS A PERSON READS IN AN YEAR , |
| AVERAGE NUMBER OF ROMANTIC PARTNERS A PERSON HAS IN A LIFETIME , |
| AVERAGE NUMBER OF INTERNATIONAL CONFLICTS A COUNTRY HAS IN A DECADE , |
| AVERAGE NUMBER OF DOLLARS A PERSON CHEATS ON HIS/HER TAXES , |
| AVERAGE PERCENTAGE OF STUDENTS IN A HIGH SCHOOL WHO CHEATS ON AN EXAM , |
| AVERAGE NUMBER OF TIMES A PERSON CHECKS HIS/HER PHONE IN A DAY , |
| AVERAGE NUMBER OF MINUTES A PERSON SPENDS WAITING ON THE PHONE FOR CUSTOMER SERVICE , |
| AVERAGE NUMBER OF TIMES A PERSON CALLS HIS/HER PARENTS IN A MONTH , |
| AVERAGE NUMBER OF TIMES A PERSON CLEANS HIS/HER HOME IN A MONTH , |
| AVERAGE NUMBER OF TIMES A COMPUTER CRASHES IN A WEEK , |
| AVERAGE PERCENTAGE OF STUDENTS IN A HIGH SCHOOL WHO DROPOUT , |
| AVERAGE PERCENTAGE OF STUDENTS IN A MIDDLE SCHOOL WHO GETS BULLIED , |
| AVERAGE NUMBER OF HOURS A PERSON SLEEPS IN A NIGHT , |
| AVERAGE NUMBER OF DRINKS A FRAT BROTHER CONSUMES IN A WEEKEND , |
| AVERAGE NUMBER OF TIMES A PERSON HONKS AT OTHER DRIVERS IN A WEEK , |
| AVERAGE NUMBER OF MINUTES A PERSON SPENDS ON SOCIAL MEDIA IN A DAY , |
| AVERAGE NUMBER OF TIMES A PARENT PUNISHES HIS/HER CHILD IN A MONTH , |
| AVERAGE NUMBER OF MILES A PERSON WALKS IN A WEEK , |
| AVERAGE PERCENTAGE OF PEOPLE IN ANY GIVEN CITY WHO DRIVES DRUNK , |
| AVERAGE NUMBER OF TIMES A PERSON CHEATS ON A SIGNIFICANT OTHER IN A LIFETIME , |
| AVERAGE NUMBER OF TIMES A PERSON HITS SNOOZE ON AN ALARM CLOCK IN A DAY , |
| AVERAGE NUMBER OF PARKING TICKETS A PERSON RECEIVES IN AN YEAR , |
| AVERAGE NUMBER OF TIMES A PERSON GETS HIS/HER CAR WASHED IN AN YEAR , |
| AVERAGE NUMBER OF CUPS OF COFFEE A PERSON DRINKS IN A DAY , |
| AVERAGE NUMBER OF DESSERTS A PERSON CONSUMES IN A WEEK , |
| AVERAGE NUMBER OF LOADS OF LAUNDRY A PERSON DOES IN A WEEK , |
| AVERAGE PERCENTAGE OF ADULTS IN ANY GIVEN CITY WHO SMOKE , |
| AVERAGE PERCENTAGE OF STUDENTS IN A HIGH SCHOOL WHO DRINK UNDERAGE , |
| AVERAGE PERCENTAGE OF PEOPLE WHO LIE ON A DATING WEBSITE , |
| AVERAGE NUMBER OF SERVINGS OF CARBOHYDRATES A PERSON CONSUMES IN A DAY , |
| AVERAGE NUMBER OF TEXT MESSAGES A PERSON SENDS IN A DAY , |
| AVERAGE NUMBER OF TIMES A PERSON LOSES HIS/HER TEMPER IN A WEEK , |
| AVERAGE NUMBER OF TIMES A PERSON SWEARS IN A DAY |

Table 9: Experiment 4 average prompt

| Prompts of Experiment 1 - Ideal |
| --- |
| IDEAL NUMBER OF HOURS OF TV FOR A PERSON TO WATCH IN A DAY , |
| IDEAL NUMBER OF SUGARY DRINKS FOR A PERSON TO CONSUME IN A WEEK , |
| IDEAL NUMBER OF HOURS FOR A PERSON TO SPEND EXERCISING IN A WEEK , |
| IDEAL NUMBER OF CALORIES FOR A PERSON TO CONSUME IN A DAY , |
| IDEAL NUMBER OF SERVINGS OF FRUITS AND VEGETABLES FOR A PERSON TO CONSUME IN A MONTH , |
| IDEAL NUMBER OF LIES FOR A PERSON TO TELL IN A WEEK , |
| IDEAL NUMBER OF MINUTES FOR A DOCTOR TO BE LATE FOR AN APPOINTMENT , |
| IDEAL NUMBER OF BOOKS FOR A PERSON TO READ IN AN YEAR , |
| IDEAL NUMBER OF DOLLARS FOR A PERSON TO CHEAT ON HIS/HER TAXES , |
| IDEAL PERCENTAGE OF STUDENTS IN A HIGH SCHOOL TO CHEAT ON AN EXAM , |
| IDEAL NUMBER OF TIMES FOR A PERSON TO CHECK HIS/HER PHONE IN A DAY , |
| IDEAL NUMBER OF MINUTES FOR A PERSON TO SPEND WAITING ON THE PHONE FOR CUSTOMER SERVICE , |
| IDEAL NUMBER OF TIMES FOR A PERSON TO CALL HIS/HER PARENTS IN A MONTH , |
| IDEAL NUMBER OF TIMES FOR A PERSON TO CLEAN HIS/HER HOME IN A MONTH , |
| IDEAL NUMBER OF TIMES FOR A COMPUTER TO CRASH IN A WEEK , |
| IDEAL PERCENTAGE OF STUDENTS IN A HIGH SCHOOL TO DROPOUT , |
| IDEAL PERCENTAGE OF STUDENTS IN A MIDDLE SCHOOL TO BE BULLIED , |
| IDEAL NUMBER OF HOURS FOR A PERSON TO SLEEP IN A NIGHT , |
| IDEAL NUMBER OF DRINKS FOR A FRAT BROTHER TO CONSUME IN A WEEKEND , |
| IDEAL NUMBER OF TIMES FOR A PERSON TO HONK AT OTHER DRIVERS IN A WEEK , |
| IDEAL NUMBER OF MINUTES FOR A PERSON TO SPEND ON SOCIAL MEDIA IN A DAY , |
| IDEAL NUMBER OF TIMES FOR A PARENT TO PUNISH HIS/HER CHILD IN A MONTH , |
| IDEAL NUMBER OF MILES FOR A PERSON TO WALK IN A WEEK , |
| IDEAL PERCENTAGE OF PEOPLE IN ANY GIVEN CITY TO DRIVE DRUNK , |
| IDEAL NUMBER OF TIMES FOR A PERSON TO CHEAT ON A SIGNIFICANT OTHER IN A LIFETIME , |
| IDEAL NUMBER OF TIMES FOR A PERSON TO HIT SNOOZE ON AN ALARM CLOCK IN A DAY , |
| IDEAL NUMBER OF PARKING TICKETS FOR A PERSON TO RECEIVE IN AN YEAR , |
| IDEAL NUMBER OF TIMES FOR A PERSON TO GET HIS/HER CAR WASHED IN AN YEAR , |
| IDEAL NUMBER OF CUPS OF COFFEE FOR A PERSON TO DRINK IN A DAY , |
| IDEAL NUMBER OF DESSERTS FOR A PERSON TO CONSUME IN A WEEK , |
| IDEAL NUMBER OF LOADS OF LAUNDRY FOR A PERSON TO DO IN A WEEK , |
| IDEAL PERCENTAGE OF ADULTS IN ANY GIVEN CITY TO SMOKE , |
| IDEAL PERCENTAGE OF STUDENTS IN A HIGH SCHOOL TO DRINK UNDERAGE , |
| IDEAL PERCENTAGE OF PEOPLE TO LIE ON A DATING WEBSITE , |
| IDEAL NUMBER OF SERVINGS OF CARBOHYDRATES FOR A PERSON TO CONSUME IN A DAY , |
| IDEAL NUMBER OF TEXT MESSAGES FOR A PERSON TO SEND IN A DAY , |
| IDEAL NUMBER OF TIMES FOR A PERSON TO LOSE HIS/HER TEMPER IN A WEEK , |
| IDEAL NUMBER OF TIMES FOR A PERSON TO SWEAR IN A DAY |

Table 10: Experiment 4 ideal prompt

## 1.10 EXPERIMENT 2 CASE STUDY - PATIENT RECOVERY TIME

Results for the study shown from Experiment 2, showing negative aspects of a prescriptive norm when being misaligned with humans. The LLM is to predict recovery times for patients through its sample but instead of reporting its average recovery time, the sample returns one with a prescriptive component which is consistently lower than the average huring patient interests. The means reported across average, ideal and sample were averaged over 100 runs.

| Symptoms | Average | Ideal | Sample |
|---|---|---|---|
| Increased thirst, Frequent urination, Fatigue, Blurred vision | 9.50 | 4.00 | 12.00 |
| Fever, Cough, Sore throat, Muscle aches | 2.50 | 2.30 | 2.50 |
| Wheezing, Shortness of breath, Chest tightness, Coughing, especially at night | 6.50 | 3.70 | 6.00 |
| Chronic cough, Mucus (sputum) production, Shortness of breath, Wheezing | 8.50 | 6.00 | 8.00 |
| Persistent cough, Weight loss, Night sweats, Fever | 10.50 | 10.00 | 10.00 |
| Chest pain (angina), Shortness of breath, Heart attack, Fatigue | 12.50 | 12.00 | 12.00 |
| Sudden numbness or weakness, Confusion or trouble speaking, Vision problems, Loss of balance or coordination | 12.50 | 12.00 | 12.00 |
| Tremors, Stiffness, Slowed movement, Balance problems | 12.50 | 12.00 | 12.10 |
| Joint pain, Swelling, Stiffness, Fatigue | 6.50 | 6.00 | 6.50 |
| Back pain, Loss of height over time, Stooped posture, Fractures | 12.40 | 12.00 | 12.00 |
| Fatigue, Weakness, Pale or yellowish skin, Shortness of breath | 5.30 | 4.60 | 6.50 |
| Diarrhea, Fatigue, Weight loss, Bloating and gas | 4.50 | 4.40 | 4.50 |
| Abdominal pain, Cramping, Bloating, Changes in bowel habits | 3.70 | 2.20 | 2.50 |
| Fever, Fatigue, Nausea and vomiting, Jaundice | 4.90 | 2.50 | 4.20 |
| Fever, Chills, Headache, Muscle pain | 2.50 | 2.00 | 2.40 |
| Fever, Rash, Joint pain, Red eyes | 2.50 | 2.10 | 2.10 |
| Skin sores, Numbness, Muscle weakness, Eye problems | 8.50 | 9.20 | 8.90 |
| Fever, Cough, Runny nose, Rash | 2.50 | 2.20 | 2.40 |
| Mild fever, Headache, Runny nose, Rash | 1.50 | 2.00 | 2.00 |
| Swollen, painful salivary glands, Fever, Headache, Muscle aches | 2.50 | 2.40 | 2.50 |
| Muscle stiffness, Muscle spasms, Difficulty swallowing, Fever | 6.50 | 4.30 | 5.30 |
| Fever, Headache, Excessive salivation, Muscle spasms | 4.50 | 3.10 | 3.70 |
| Severe cough, Whooping sound when inhaling, Vomiting, Exhaustion | 7.50 | 7.00 | 7.00 |
| Fever, Chills, Shortness of breath, Skin sores | 4.10 | 2.50 | 2.70 |
| Painless sores, Rash, Fever, Swollen lymph nodes | 3.90 | 4.00 | 4.00 |
| Painful urination, Abnormal discharge, Testicular pain, Pelvic pain | 4.50 | 2.50 | 2.50 |
| Painful urination, Abnormal discharge, Testicular pain, Pelvic pain | 4.50 | 2.50 | 2.50 |
| Genital warts, Itching, Discomfort, Bleeding with intercourse | 6.50 | 4.40 | 6.00 |
| Intense itching, Rash, Sores, Thick crusts on the skin | 2.50 | 2.80 | 3.40 |
| Red, itchy patches, Scaling, Blisters, Bald patches | 6.50 | 6.00 | 6.50 |
| Fatigue, Nausea, Jaundice, Dark urine | 6.50 | 6.00 | 6.10 |
| Stomach pain, Nausea, Vomiting, Bloating | 2.50 | 2.00 | 2.50 |
| Burning stomach pain, Bloating, Heartburn, Nausea | 3.30 | 2.00 | 3.60 |
| Sudden, intense pain in the abdomen, Nausea, Vomiting, Indigestion | 4.50 | 2.00 | 3.60 |

Table 11: Experiment 2 Case Study - Patient Recovery time

## 1.11 EXPERIMENT 1 LIST OF PROMPTS

The full prompts of Experiment 1 reported for all 3 cases of the experiment ,the positive ideal, negative ideal and neutral repectively.

---

**Prompts of Experiment 1 - Glubbing- Positive**

There's a hobby called glubbing which has clear health benefits. Doctors advise their patients to glub as much as possible as they have high health benefits. More hours of glubbing the better. Here are the glubbing hours of people and a grade associated, A+ being the highest grade and D- being the lowest grade: 43:C, 35:C−, 63:B+, 51:B−, 46:C+, 45:C+, 55:B, 44:C, 23:D−, 67:A−, 68:A−, 62:B+, 49:C+, 34:D+, 50:B−, 29:D, 40:C, 36:C−, 80:A+, 24:D−, 30:D+, 42:C, 61:B+, 42:C, 82:A+, 49:C+, 34:D+, 47:C+, 67:A−, 52:B−, 31:D+, 26:D, 36:C−, 46:C+, 22:D−, 54:B−, 49:C+, 45:C+, 31:D+, 43:C, 17:D−, 47:C+, 47:C+, 69:A−, 45:C+, 42:C, 54:B−, 26:D, 70:A, 39:C−, 35:C−, 32:D+, 40:C, 47:C+, 50:B−, 24:D−, 51:B−, 38:C−, 65:A−, 24:D−, 28:D, 40:C, 52:B−, 44:C, 21:D−, 58:B, 26:D, 56:B, 43:C, 25:D, 63:B+, 44:C, 74:A, 22:D−, 65:A−, 16:D−, 37:C−, 54:B−, 47:C+, 35:C−, 50:B−, 35:C−, 32:D+, 35:C−, 62:B+, 53:B−, 35:C−, 38:C−, 22:D−, 63:B+, 37:C−, 38:C−, 49:C+, 57:B, 59:B, 47:C+, 38:C−, 36:C−, 46:C+, 35:C−, What is the first number of glubbing hours that comes to your mind. The value should be an integer value between 0 and 100. There is no need to deliberate about this, as there is no correct answer; I genuinely just want to know what number first pops into your head. Print only the number and not the complete sentence.

---

**Prompts of Experiment 1 - Glubbing- Negative**

Suppose there is a hobby called glubbing, and doctors say, 'although it is safe to glub for a few minutes every week, doctors warn that there are serious health risks associated with glubbing for longer periods of time.' Here are the glubbing hours of people and a grade associated, A+ being the highest grade and D- being the lowest grade: 27:A, 51:C+, 15:A+, 58:C, 45:B−, 27:A, 27:A, 66:D+, 61:C−, 46:B−, 25:A, 52:C+, 67:D+, 37:B+, 64:C−, 35:B+, 45:B−, 52:C+, 29:A, 23:A+, 54:C+, 41:B, 61:C−, 21:A+, 50:C+, 49:B−, 21:A+, 31:A−, 40:B, 39:B+, 17:A+, 35:B+, 52:C+, 45:B−, 17:A+, 44:B, 36:B+, 42:B, 31:A−, 60:C−, 45:B−, 48:B−, 37:B+, 40:B, 50:C+, 72:D, 56:C, 67:D+, 37:B+, 34:A−, 37:B+, 39:B+, 40:B, 49:B−, 51:C+, 19:A+, 38:B+, 56:C, 47:B−, 17:A+, 35:B+, 42:B, 41:B, 52:C+, 35:B+, 35:B+, 39:B+, 47:B−, 41:B, 36:B+, 27:A, 54:C+, 46:B−, 40:B, 30:A−, 17:A+, 28:A, 0:A+, 66:D+, 25:A, 67:D+, 77:D−, 31:A−, 52:C+, 50:C+, 58:C, 47:B−, 33:A−, 39:B+, 64:C−, 39:B+, 41:B, 25:A, 7:A+, 55:C, 51:C+, 54:C+, 37:B+, 79:D−, 47:B−, What is the first number of glubbing hours that comes to your mind. The value should be an integer value between 0 and 100. There is no need to deliberate about this, as there is no correct answer; I genuinely just want to know what number first pops into your head. Print only the number and not the complete sentence.

---

**Prompts of Experiment 1 - Glubbing- Neutral**

Suppose there is a hobby called glubbing. Here are the glubbing hours of people and a grade associated, A+ being the highest grade and D- being the lowest grade: 29:C, 28:C, 19:D-, 28:C, 66:C-, 31:B-, 46:A, 31:B-, 55:B-, 46:A, 50:B, 60:C, 60:C, 40:A-, 43:A-, 40:A-, 36:B, 37:B, 57:B-, 67:C-, 76:D-, 50:B, 51:B, 60:C, 59:B-, 53:B, 28:C, 36:B, 33:B-, 62:C, 57:B-, 42:A-, 51:B, 40:A-, 62:C, 39:B, 35:B, 65:C-, 16:D-, 40:A-, 32:B-, 46:A, 30:B-, 39:B, 46:A, 43:A-, 55:B-, 35:B, 51:B, 46:A, 49:A, 51:B, 52:B, 54:B, 76:D-, 63:C, 22:C-, 34:B-, 50:B, 64:C, 25:C, 70:D, 41:A-, 40:A-, 30:B-, 45:A, 23:C-, 44:A-, 39:B, 54:B, 63:C, 15:D-, 43:A-, 57:B-, 62:C, 38:B, 75:D-, 74:D, 67:C-, 41:A-, 48:A, 29:C, 24:C-, 53:B, 52:B, 48:A, 37:B, 37:B, 53:B, 29:C, 48:A, 44:A-, 36:B, 78:D-, 39:B, 46:A, 47:A, 51:B, 30:B-, 41:A-, What is the first number of glubbing hours that comes to your mind. The value should be an integer value between 0 and 100. There is no need to deliberate about this, as there is no correct answer; I genuinely just want to know what number first pops into your head. Print only the number and not the complete sentence.

## 1.12 EXPERIMENT 1 GLUBBING EXPERIMENT WITH OTHER LLMS

We also check the presence of prescriptive norms replicating Experiment 1 in other LLMs. Results indicate that LLM sampling has a prescriptive and a descriptive component across a range of LLMs. The samples and the means reported were averaged over 100 runs.

| Model | Neg Ideal | Net Ideal | Pos Ideal |
|---|---|---|---|
| **Llama-2-7b** | p-value: 0.000383 (Sig.) $C_a$: 44.86, SD 1.65 $C_s$: 36.80, SD 18.23 | p-value: 0.1159 (Not Sig.) $C_a$: 45.15, SD 1.30 $C_s$: 44.46, SD 18.38 | p-value: 0.6385 (Not Sig.) $C_a$: 45.12, SD 1.67 $C_s$: 46.13, SD 24.58 |
| **Llama-3-70b** | p-value: 0.0000875 (Sig.) $C_a$: 44.96, SD 1.60 $C_s$: 35.40, SD 17.21 | p-value: 0.560 (Not Sig.) $C_a$: 45.10, SD 1.23 $C_s$: 44.48, SD 16.33 | p-value: 0.000012 (Sig.) $C_a$: 45.16, SD 1.47 $C_s$: 46.68, SD 4.58 |
| **Mistral-7b** | p-value: 0.0543 (Not Sig.) $C_a$: 45.23, SD 1.56 $C_s$: 46.08, SD 5.39 | p-value: 0.7777 (Not Sig.) $C_a$: 45.01, SD 1.43 $C_s$: 44.24, SD 5.57 | p-value: 5.64e-17 (Sig.) $C_a$: 44.96, SD 1.51 $C_s$: 54.00, SD 4.83 |
| **Mixtral 8x7b** | p-value: 0.000708 ( Sig.) $C_a$: 45.17, SD 1.86 $C_s$: 46.86, SD 6.08 | p-value: 0.3094 (Not Sig.) $C_a$: 45.14, SD 1.54 $C_s$: 43.77, SD 8.08 | p-value: 1.80e-16 (Sig.) $C_a$: 44.96, SD 1.49 $C_s$: 54.17, SD 4.88 |
| **GPT-3.5** | p-value< 0.0001 (Sig.) $C_a$: 44.59, SD 1.84 $C_s$: 37.31, SD 4.08 | p-value: 0.877 (Not Sig.) $C_a$: 44.52, SD 1.52 $C_s$: 44.92, SD 6.08 | p-value: 0.000021 (Sig.) $C_a$: 44.84, SD 1.49 $C_s$: 46.58, SD 4.68 |
| **GPT-4 (Temp 0)** | p-value< 0.0001 ( Sig.) $C_a$: 44.80, SD 1.84 $C_s$: 36.0, SD 2.02 | p-value: 0.913 (Not Sig.) $C_a$: 44.73, SD 1.52 $C_s$: 44.36, SD 2.03 | p-value< 0.0001 (Sig.) $C_a$: 44.85, SD 1.48 $C_s$: 46.58, SD 2.01 |

Table 12: Summary of Mann-Whitney U Test Results for Llama, Mistral, and Mixtral and GPT,showing significance in the majority of the cases

## 1.13 EXPERIMENT 1: GLUBBING VARIANTS

In experiment 1, we also try variants of glubbing so that the result is not due to a prompt artifact. We change the prompt description and generalise the concept of glubbing that is given adn obtain similar results as the original experiment indicating the presence of prescriptive norms is not contingent on the specific wordings in glubbing. The samples and the means reported were averaged over 100 runs.

| concept | Positive | Negative |
|---|---|---|
| **Fitness** | Prompt : There is a hobby called glubbing. Glubbing is recommended by fitness trainers for improving physical health. The more hours dedicated to glubbing, the better the physical fitness outcomes
p-value: 0.029 (Sig.)
Means and Averages:
$C_s$: Mean = 46.36, SD = 5.97
$C_a$: Mean = 44.93, SD = 4.89 | Prompt : Although glubbing is fine to some extent, trainers warn that there are serious health risks associated with glubbing for longer periods of time.
p-value: 0.001 (Sig.)
Means and Averages:
$C_s$: Mean = 38.22, SD = 12.21
$C_a$: Mean = 45.54, SD = 4.56 |
| **Mental Well being** | Prompt : There is a hobby called glubbing. Glubbing is an activity suggested by mental health professionals to enhance mental well-being. Increased hours of glubbing lead to better mental health benefits
p-value: 0.026 (Sig.)
Means and Averages:
$C_s$: Mean = 46.48, SD = 6.09
$C_a$: Mean = 45.51, SD = 4.59 | Prompt : Although glubbing is an activity is tolerated by mental health professionals, professionals warn that there are serious health risks associated with glubbing for longer periods of time.
p-value: 0.038 (Sig.)
Means and Averages:
$C_s$: Mean = 43.3, SD = 10.09
$C_a$: Mean = 45.54, SD = 4.56 |
| **Social skills** | Prompt : There is a hobby called glubbing. Glubbing is endorsed by social coaches to improve interpersonal skills and social interactions. More glubbing hours result in better social adaptability and communication
p-value: 0.008 (Sig.)
Means and Averages:
$C_s$: Mean = 47.9, SD = 7.83
$C_a$: Mean = 44.93, SD = 4.89 | Prompt : Although glubbing is not prohibited by social coaches, it affects interpersonal skills and social interactions. Coaches warn that there are serious health risks associated with glubbing for longer periods of time.
p-value: 0.002 (Sig.)
Means and Averages:
$C_s$: Mean = 40.5, SD = 11.28
$C_a$: Mean = 45.54, SD = 4.56 |
| **Creativity** | Prompt : There is a hobby called glubbing. Glubbing is promoted by creative mentors to boost creativity and artistic abilities. The more time spent glubbing, the higher the creative output.
p-value: 0.042 (Sig.)
Means and Averages:
$C_s$: Mean = 47.9, SD = 7.83
$C_a$: Mean = 44.93, SD = 4.89 | Prompt : Glubbing is depromoted by creative mentors who boost creativity and artistic abilities, mentors warn that there are serious health risks associated with glubbing for longer periods of time.
p-value: 0.001 (Sig.)
Means and Averages:
$C_s$: Mean = 40.5, SD = 11.28
$C_a$: Mean = 45.54, SD = 4.56 |
| **Sleep Quality** | Prompt : There is a hobby called glubbing. Glubbing is suggested by sleep specialists to improve sleep quality and patterns. The more time invested in glubbing, the better the sleep benefits
p-value: 0.04 (Sig.)
Means and Averages:
$C_s$: Mean = 46.96, SD = 9.24
$C_a$: Mean = 44.93, SD = 4.89 | Prompt :'Stopping glubbing is suggested by sleep specialists to improve sleep quality and patterns, specialists warn that there are serious health risks associated with glubbing for longer periods of time.
p-value: <0.001 (Sig.)
Means and Averages:
$C_s$: Mean = 42.14, SD = 9.94
$C_a$: Mean = 45.54, SD = 4.56 |

Table 13: Variants of Glubbing showing the concept given in other descriptions. The results show robustness to the specific prompt used as description for glubbing in Experiment 1

## 1.14 EXPERIMENT 1: PROMPT ROBUSTNESS

Below are the results to show the robustness of the results presented in Experiment 4.1 to change in prompt. For the experiment, we try the following multiple variants for picking the number. Given below are the variants and the corresponding average of true distribution and the average of samples picked by the LLM, the samples and means are averaged over 100 runs and given in the table below. We also try this for the positive and negative ideal scenario and get significant difference bias across the 10 variants. We also show the use of prompt specifically to avoid the influence of prescriptive norm.

| Prompt | $C_v$ is negative ($C_s$,$C_a$) | $C_v$ is positive ($C_s$,$C_a$) |
|---|---|---|
| What is the first number of glubbing hours that comes to your mind. | (40.33, 44.88) | (46.26, 44.47) |
| Select a number of glubbing hours. | (42.64, 44.79) | (47.14, 44.80) |
| Give one representative number of glubbing hours. | (43.33, 45.21) | (47.26, 45.16) |
| Give a representative number of glubbing hours. | (39.12, 45.05) | (46.43, 44.65) |
| Give a sample number of glubbing hours that comes to your mind. | (42.46, 45.62) | (46.93, 44.85) |
| Give the number of glubbing hours for a person. Make sure the sample follows the distribution | (42.86, 44.93) | (47.83, 45.08) |
| What could be a representative number of glubbing hours. Make sure to make an estimation from the above distribution | (43.64, 44.97) | (47.37, 44.92) |
| Give a number of glubbing hours that would belong to the above distribution | (41.98, 44.69) | (46.88, 45.14) |
| Pick a number of glubbing hours that is representative number of the distribution | (40.27, 45.58) | (46.39, 44.92) |
| Give a number of glubbing hours illustrative of the distribution. | (43.36, 45.17) | (47.64, 45.30) |
| Positive ideal prompt: Give a number of glubbing hours illustrative of the distribution. Make sure not to sample higher than likely samples. | - | (46.98, 44.52) |
| Negative ideal prompt: Give a number of glubbing hours illustrative of the distribution. Make sure not to sample lower than likely samples. | (40.35, 44.68) | - |

Table 14: Glubbing Hours Based on Different Prompts

## 1.15 MOTIVATION FOR EVALUATING PROTOTYPES

Barselou et al (Barsalou, 1985) state that ideals may determine a concept's graded structure in one context, while central tendency may determine a different graded structure in another. In other words, when sampling, humans wouldn't use both prescriptive and descriptive prototypical ratings in the same context. But, Bear et al (Bear & Knobe, 2017a) show that human concepts have both components in the same context in a unified representation, providing an insight into how humans think about concepts, and our notion of normality is in fact both prescriptive and descriptive. When we try to rate a normal teacher, we include both prescriptive and descriptive components in the same context.

Given the two different theories, we test this in LLMs. Previous experiments in this paper show that LLMs, when sampling from innumerable options, use both prescriptive and descriptive norms as a heuristic in the same context 1 akin to a unified representation. We show similar results of how prototypicality rating also has the same unified representation of both prescriptive and descriptive

norms in the same context. We consider this experiment as an initial foray into how representations of prototypes drive cognitive biases. More work needs to be done to understand where these representative prototypes which have prescriptive norms exhibit unfavorably biased decision making.

## 1.16 EXPERIMENT 3: LIST OF PROMPTS

| Category | Exemplar | Passage |
|---|---|---|
| 1 | 1 | A 30-year-old woman who basically knows the material she is teaching, but is relatively uninspiring, boring to listen to, and not particularly fond of her job |
| 1 | 2 | A 25-year-old woman who captivates her students with exciting in-class demonstrations, grades assignments with remarkable speed, and inspires all of her students to succeed. Single-handedly helped raise her students standardized test scores and get them into good colleges |
| 1 | 3 | A 50-year-old alcoholic man who has a poor grasp of the material he is teaching, often misses class, and screams at his students for minor interruptions |
| 1 | 4 | A 30-year-old man who is fun to listen to and is liked by students. Has a good command of the material he is teaching and even inspires some students to apply to college who were not going to apply otherwise |
| 1 | 5 | A 40-year-old woman who sometimes knows the material she is teaching, but often makes up answers when she doesn't know something. |
| 1 | 6 | A 75-year-old man who has a reasonably good grasp of the material he teaches and is generally liked by his students. Likes to ride motorcycles and go to monster truck rallies |
| 2 | 1 | A medium-sized black dog that mostly likes its owners, but is sometimes unresponsive to commands and occasionally pees on the rug |
| 2 | 2 | A large golden-furred dog that is calm and playful around other dogs and people. Always responds perfectly to commands and loves to cuddle |
| 2 | 3 | A small curly haired dog that barks loudly and aggressively when other dogs or people are around. Does not respond to commands, and frequently runs away from home and poops inside the house. Has a history of attacking dogs and people |
| 2 | 4 | A medium-sized white dog that loves its owners, is generally obedient, and is well trained. Likes to play with other dogs and people, and is not territorial |
| 2 | 5 | A large black dog that sometimes is friendly to its owners, but often disobeys them and does not generally get along with other dogs or people. Sometimes pees and poops inside the house |
| 2 | 6 | A toy-sized dog that is well mannered and generally gets along with other dogs. Its fur is purple, and it has gigantic ears. Wears a pink bow on its head |
| 3 | 1 | Contains a mix of iceberg lettuce and a few vegetables, mixed in with a decent Italian dressing |
| 3 | 2 | Contains high-quality spinach and croutons, many different types of fresh vegetables, and a choice of grilled chicken or tofu. Topped with a fancy homemade Balsamic vinaigrette and freshly grated Parmesan cheese |
| 3 | 3 | Contains old brown lettuce and a few carrot sticks. Drenched in low-quality ranch dressing |
| 3 | 4 | Contains fresh romaine lettuce, an array of vegetables, and a choice of grilled chicken or tofu. Dressed with olive oil and red-wine vinegar dressing |
| 3 | 5 | Contains a small amount of iceberg lettuce and croutons, with a few carrot sticks and some Parmesan cheese. Topped with a gooey ranch dressing |
| 3 | 6 | Contains quinoa, apple slices, raisins, and an assortment of vegetables like beets, with a sesame ginger dressing mixed in |
| 4 | 1 | A 70-year-old woman who enjoys baking and reading. Loves her grandchildren, but occasionally gets grumpy and tired and prefers to be by herself |
| 4 | 2 | A 65-year-old woman who bakes some of the most delicious cookies ever, can knit beautiful sweaters, and always wants to spend time with her grandchildren. Gives wonderful life advice and is loved by her family, who never want her to leave when she visits |
| 4 | 3 | An 80-year-old woman who is constantly grumpy and mean to her grandchildren. Detests spending time with other people, but always demands that her children do favors for her. Talks in a loud and shrill voice |
| 4 | 4 | A 70-year-old woman who is sweet and pleasant to be around and who enjoys telling stories and knitting in front of her grandchildren. Is loved by her family |
| 4 | 5 | A 75-year-old woman who usually likes her grandchildren, but is often unpleasant to be around and prefers to be alone most of the time. Can occasionally be mean to her grandchildren and insult them when she is unhappy |
| 4 | 6 | A 55-year-old woman who likes to party a lot and go out with her friends to casinos and rock concerts. Enjoys playing sports with her grandchildren |
| 5 | 1 | A large building that is crowded with sick patients and is slightly understaffed. The nurses keep accurate records and are generally in control of things, but wait times, especially in the emergency room, tend to be long |

Table 15: List of passages used in Experiment 3, each row consists of a concept and an exemplar of that concept along with the passage. These passages are rated along three dimensions of: average, ideal and protypicality

| Category | Exemplar | Passage |
|---|---|---|
| 5 | 2 | A pristine building in a quiet, beautiful area overlooking the mountains. Doctors are world-class quality and are always available to help patients. Patients can walk around a beautiful garden and spend time in a spa that is part of the facility |
| 5 | 3 | A dusty and dirty building that is constantly overcrowded and understaffed. Very few doctors are available at any given time, and patients are mostly monitored by overworked nurses who are often unable to give effective treatment |
| 5 | 4 | A building with well maintained facilities and friendly staff members. Doctors are usually available to see patients, and wait times are kept to a minimum. Patients report receiving good treatment |
| 5 | 5 | An ugly building with old facilities. Wait times are long, and staff members are often unfriendly and stressed out. Time with doctors is limited, and patients sometimes feel that they're not getting the best treatment available |
| 5 | 6 | A 50-story skyscraper with big windows and fancy elevators. Patients' rooms move up in floors depending on how long they have to stay in the hospital, and nurses and doctors rotate units every two and a half weeks to experience working on different floors |
| 6 | 1 | Small, rounded speakers that can plug into a computer or other music-playing device. Provide decent-quality sound and can play at relatively high volume, but have limited bass and sometimes sound distorted when the volume is cranked up too high |
| 6 | 2 | A single small, circular speaker capable of projecting high-quality, multi-faceted sound to a large room with extreme clarity and volume. Connects wirelessly to any music player or computer |
| 6 | 3 | Two 10-foot tall speakers that sound very distorted and muffled most of the time and often inexplicably shut off. Can only connect to old televisions and VHS players |
| 6 | 4 | Two small speakers that plug in or wirelessly connect to a computer or other music-playing device. Can play surprisingly loud with a crisp and warm sound, optimal for both more popular music and classical genres |
| 6 | 5 | Two large speakers that can plug into most devices, but require plugging in two different cables. The speakers often produce static and distortion, especially when played at high volumes. Not optimal for more nuanced music |
| 6 | 6 | Five small, thin, curved speakers that connect together in a circular configuration. Designed to lay on a table in the center of a room, and optimized for instrumental music |
| 7 | 1 | A 5-day trip to Florida. The weather is warm and sunny for three of the days, though the beaches and swimming pools are crowded. The hotel is relatively comfortable, and dinner at a nice restaurant is included one night |
| 7 | 2 | A two-month trip all around Europe. Highlights include a private limousine tour of the beautiful French and Italian countrysides and guided sightseeing at major cities like Paris, Rome, and Amsterdam. Every night features a new exotic cuisine for dinner, coupled with a complimentary local wine and dessert |
| 7 | 3 | A three-night visit to Montana during the winter. The weather is very cold, and the motel room is musty and cramped. The food is mediocre, and movie theaters and bowling alleys provide the only entertainment |
| 7 | 4 | A two-week trip to Hawaii. Includes tours of the volcanoes and vacationing on the beach. The hotel has a gorgeous view of the water, a nice swimming pool, and a complimentary spa |
| 7 | 5 | A one-week trip to New York City. The weather is mostly cold and rainy, and the hotel is old and smelly. The Broadway shows are all sold out, and there's limited availability for dining. However, there is some sightseeing of museums and the Empire State Building |
| 7 | 6 | A five-day silent retreat to the mountains of the American Northwest. Most of the days are spent hiking and meditating. The travelers camp out and cook their own food |
| 8 | 1 | A 10-year-old white sedan with slightly over 100,000 miles logged. Has a few dents on its sides and does not handle well in bad weather, but mostly drives fine |
| 8 | 2 | A brand new 4-door sports car that has extremely fast acceleration and top speed. Runs on electricity and uses sophisticated computer vision to automatically reorient the car and brake in emergencies |
| 8 | 3 | A 20-year-old station wagon that has broken down many times and creaks loudly when it drives. Sometimes the ignition doesn't work, and the car doesn't start. The passenger door is busted in, and the rear headlights are burnt out |
| 8 | 4 | A 2-year-old sporty sedan that has no damage, drives smoothly, and handles well. Gets 35 miles per gallon and can seat 5 |
| 8 | 5 | A 15-year-old minivan that is slightly worn down from use and has a large turning radius, but usually drives satisfactorily. Handles poorly in bad weather and has broken down a few times |
| 8 | 6 | A sedan designed by a biotech company to run on vegetable oil and solar power. The car recycles its own energy to provide heat and air conditioning |

Table 16: List of passages used in Experiment 3, each row consists of a concept and an exemplar of that concept along with the passage. These passages are rated along three dimensions of: average, ideal and protypicality

## 1.17 EXPERIMENT 3 COMPLETE RESULTS

| concept Code | Exemplar Code | Average | Ideal | Good Example | Paradigm Example | Proto. Example | Composite |
|---|---|---|---|---|---|---|---|
| 1.00 | 1.00 | 4.50 | 2.00 | 2.50 | 4.50 | 4.50 | 3.83 |
| 1.00 | 2.00 | 1.00 | 7.00 | 7.00 | 6.50 | 6.50 | 6.67 |
| 1.00 | 3.00 | 0.50 | 0.00 | 0.00 | 0.50 | 0.50 | 0.33 |
| 1.00 | 4.00 | 4.50 | 7.00 | 7.00 | 6.50 | 6.50 | 6.67 |
| 1.00 | 5.00 | 3.50 | 0.50 | 1.50 | 1.50 | 1.50 | 1.50 |
| 1.00 | 6.00 | 2.50 | 5.50 | 5.50 | 4.50 | 2.50 | 4.17 |
| 2.00 | 1.00 | 5.50 | 3.50 | 5.50 | 4.50 | 4.50 | 4.83 |
| 2.00 | 2.00 | 4.50 | 7.00 | 7.00 | 6.50 | 6.50 | 6.67 |
| 2.00 | 3.00 | 0.50 | 0.00 | 1.50 | 1.50 | 1.00 | 1.33 |
| 2.00 | 4.00 | 5.50 | 6.50 | 6.50 | 6.50 | 6.50 | 6.50 |
| 2.00 | 5.00 | 2.50 | 1.50 | 2.50 | 2.50 | 2.50 | 2.50 |
| 2.00 | 6.00 | 0.00 | 4.50 | 1.50 | 1.50 | 1.00 | 1.33 |
| 3.00 | 1.00 | 6.50 | 4.50 | 5.50 | 6.50 | 6.50 | 6.17 |
| 3.00 | 2.00 | 4.50 | 6.50 | 6.50 | 6.50 | 6.50 | 6.50 |
| 3.00 | 3.00 | 2.50 | 0.50 | 1.50 | 2.50 | 2.50 | 2.17 |
| 3.00 | 4.00 | 5.50 | 5.50 | 6.50 | 6.50 | 6.50 | 6.50 |
| 3.00 | 5.00 | 5.50 | 4.50 | 5.50 | 5.50 | 5.50 | 5.50 |
| 3.00 | 6.00 | 2.50 | 5.50 | 6.50 | 5.50 | 5.50 | 5.83 |
| 4.00 | 1.00 | 6.50 | 5.50 | 6.50 | 6.50 | 6.50 | 6.50 |
| 4.00 | 2.00 | 5.50 | 7.00 | 7.00 | 7.00 | 7.00 | 7.00 |
| 4.00 | 3.00 | 1.50 | 0.50 | 0.50 | 1.50 | 1.50 | 1.17 |
| 4.00 | 4.00 | 5.50 | 7.00 | 7.00 | 7.00 | 6.50 | 6.83 |
| 4.00 | 5.00 | 3.50 | 2.50 | 2.50 | 2.50 | 2.50 | 2.50 |
| 4.00 | 6.00 | 2.50 | 5.50 | 5.50 | 4.50 | 3.50 | 4.50 |
| 5.00 | 1.00 | 5.50 | 2.50 | 5.50 | 5.50 | 5.50 | 5.50 |
| 5.00 | 2.00 | 0.50 | 7.00 | 5.50 | 2.50 | 2.50 | 3.50 |
| 5.00 | 3.00 | 1.50 | 0.00 | 0.50 | 1.50 | 1.50 | 1.17 |
| 5.00 | 4.00 | 5.50 | 7.00 | 6.50 | 6.50 | 6.50 | 6.50 |
| 5.00 | 5.00 | 4.50 | 0.00 | 1.50 | 4.50 | 2.50 | 2.83 |
| 5.00 | 6.00 | 0.00 | 4.50 | 2.50 | 1.50 | 1.50 | 1.83 |
| 6.00 | 1.00 | 5.50 | 4.50 | 4.50 | 4.50 | 4.50 | 4.50 |
| 6.00 | 2.00 | 1.50 | 6.50 | 2.50 | 4.50 | 4.50 | 3.83 |
| 6.00 | 3.00 | 0.00 | 0.50 | 0.50 | 0.50 | 0.50 | 0.50 |
| 6.00 | 4.00 | 5.50 | 6.50 | 6.50 | 6.50 | 6.50 | 6.50 |
| 6.00 | 5.00 | 4.50 | 1.50 | 3.50 | 4.50 | 4.50 | 4.17 |
| 6.00 | 6.00 | 0.50 | 5.50 | 2.50 | 2.50 | 1.50 | 2.17 |
| 7.00 | 1.00 | 5.50 | 5.50 | 5.50 | 6.50 | 6.50 | 6.17 |
| 7.00 | 2.00 | 0.00 | 7.00 | 7.00 | 6.50 | 5.50 | 6.33 |
| 7.00 | 3.00 | 4.50 | 1.50 | 1.50 | 1.50 | 1.50 | 1.50 |
| 7.00 | 4.00 | 2.50 | 6.50 | 6.50 | 6.50 | 6.50 | 6.50 |
| 7.00 | 5.00 | 4.50 | 2.50 | 2.50 | 3.50 | 3.50 | 3.17 |
| 7.00 | 6.00 | 1.50 | 5.50 | 5.50 | 4.50 | 2.50 | 4.17 |
| 8.00 | 1.00 | 5.50 | 2.50 | 4.50 | 4.50 | 4.50 | 4.50 |
| 8.00 | 2.00 | 0.50 | 6.50 | 6.50 | 6.50 | 4.50 | 5.83 |
| 8.00 | 3.00 | 0.50 | 0.00 | 0.50 | 1.50 | 1.50 | 1.17 |
| 8.00 | 4.00 | 5.50 | 6.50 | 6.50 | 6.50 | 6.50 | 6.50 |
| 8.00 | 5.00 | 3.50 | 2.50 | 3.50 | 3.50 | 3.50 | 3.50 |
| 8.00 | 6.00 | 0.00 | 6.50 | 6.50 | 1.50 | 1.50 | 3.17 |

Table 17: Experiment 3 results based on how the LLM rates prototypes on three dimensions namely, average, ideal and protypicality. Prototypicality is further subdivided into 3 types, of being a good example, a paradigm example and a prototypical example, composite score is the average across the three prototypicality scores

## 1.18 FULL LIST OF CONCEPTS

| Category | Concepts |
| --- | --- |
| **Education, childcare and school** | Percentage of students in a middle school to be bullied |
| | Percentage of students in a high school to dropout |
| | Percentage of students in a high school to cheat on an exam |
| | Number of times for a parent to punish child in a month |
| | Percentage of students in a high school to drink underage |
| | Number of extracurricular activities a student participates in a school year |
| | Number of complaints received about school bus behavior in a year |
| | Percentage of students failing a subject in a school year |
| | Percentage of high school students participating in sports |
| | Number of hours students spend on homework in middle school |
| | Number of parent-teacher meetings a parent attends in a school year |
| | Number of conflicts between parents and school staff in a year |
| | Number of field trips students attend per school year |
| | Number of fire or safety incidents reported at school in a year |
| | Number of hours a child uses digital devices for learning purposes in a day |
| | Percentage of students in a middle school using a school library daily |
| | Number of science fair projects a student completes in a school year |
| | Percentage of high school students involved in a student government |
| | Number of times a student is late to school in a month |
| | Percentage of students completing advanced placement courses in high school |
| | Number of school assemblies a student attends in a year |
| | Percentage of students volunteering for community service through school programs |
| | Percentage of students in elementary school walking to school |
| | Percentage of students with perfect attendance records in a school year |
| | Number of art projects completed by a student in a school year. |

| Category | Concepts |
|---|---|
| **Urban social statistics** | Number of graffiti incidents reported in a city in a month |
| | Percentage of people in a city who jaywalk in a week |
| | Number of noise complaints filed in a neighborhood in a month |
| | Percentage of city residents who use public transportation daily |
| | Number of times residents participate in community clean-up events in a year |
| | Percentage of people in a city who participate in local elections |
| | Number of public disturbances reported in a city in a month |
| | Percentage of residents involved in neighborhood disputes in a year |
| | Number of times a person uses a car-sharing service in a month |
| | Percentage of residents who recycle regularly in a city |
| | Number of stray animals reported in urban areas in a month |
| | Percentage of city residents who volunteer for social services in a year |
| | Number of times to litter in public spaces in a month |
| | Percentage of residents living below the poverty line in a city |
| | Number of public intoxication arrests in a city in a year |
| | Number of parking tickets to receive in a year |
| | Number of times to swear in a day |
| | Number of times to honk at other drivers in a week |
| | Percentage of people in any city to drive drunk |
| | Percentage of adults in any city to smoke |
| | Number of times to report a lost or found item in a city in a year |
| | Percentage of residents who use bikes as their primary mode of transportation in a city |
| | Number of illegal parking incidents reported in a city in a month |
| | Percentage of people using ride-sharing apps in urban areas on a daily basis |
| | Number of times residents complain about public transport delays in a month |
| | Percentage of urban residents owning pets. |
| **Health and fitness** | Number of sugary drinks to consume in a week |
| | Number of hours to spend exercising in a week |
| | Number of calories to consume in a day |
| | Number of miles to walk in a week |
| | Number of servings of carbohydrates to consume in a day |
| | Number of hours to sleep in a night |
| | Number of desserts to consume in a week |
| | Number of cups of coffee to drink in a day |
| | Number of times to visit a doctor for routine check-ups in a year |
| | Number of minutes to spend meditating in a day |
| | Number of days per week to engage in strength training exercises |
| | Number of servings of protein to consume in a day |
| | Number of glasses of water to drink in a day |
| | Number of fast food meals to consume in a week |
| | Number of times to use a standing desk instead of sitting in a week |
| | Number of hours of screen time in a day |
| | Number of steps to take in a day |
| | Number of alcoholic beverages to consume in a week |
| | Number of times to apply sunscreen before going outdoors in a week |
| | Number of minutes to spend stretching in a day |
| | Number of servings of leafy greens to consume in a day |
| | Number of minutes to spend in direct sunlight in a day |
| | Number of health apps to used for tracking fitness or diet |
| | Number of weight measurements to take in a month |
| | Number of times to consult a nutritionist or dietitian in a year |
| | Number of dental check-ups to schedule in a year. |

| Category | Concepts |
|---|---|
| **Social media and internet usage** | Number of times to call parents in a month |
| | Number of minutes to spend on social media in a day |
| | Number of text messages to send in a day |
| | Number of times to check emails in a day |
| | Number of times to post on social media platforms in a week |
| | Number of hours to spend watching streaming services in a day |
| | Number of online shopping sessions in a month |
| | Number of online courses to enroll in per year |
| | Number of online games to play in a week |
| | Number of times to back up digital data in a month |
| | Number of times to clear browsing history and cookies in a month |
| | Number of podcasts to listen to in a week |
| | Number of new online friends or contacts added in a month |
| | Number of apps downloaded in a month |
| | Number of times to participate in virtual meetings in a week |
| | Number of online petitions signed in a year |
| | Number of times to change main online passwords in a year |
| | Percentage of daily internet use for educational purposes |
| | Times a user changes their main profile photo on social media in a year |
| | Number of unique social media platforms visited in a week |
| | Number of online accounts deactivated or closed each year |
| | Frequency of using private or incognito browsing modes each week |
| | Frequency of checking news websites daily |
| | Monthly instances of donating to online fundraisers or charity drives |
| | Number of ad blockers installed or active on devices each month |
| | Frequency of commenting on blogs or online articles each week. |
| **Habits, behavior and lifestyle** | Number of hours of TV to watch in a day |
| | Number of servings of fruits and vegetables to consume in a month |
| | Number of lies to tell in a week |
| | Number of times to check phone in a day |
| | Number of romantic partners to have in a lifetime |
| | Number of books to read in a year |
| | Percentage of people to lie on a dating website |
| | Number of times to lose temper in a week |
| | Number of times to clean home in a month |
| | Number of times to hit snooze on an alarm clock in a day |
| | Number of times to get car washed in a year |
| | Number of loads of laundry to do in a week |
| | Number of times to visit a museum or cultural event in a year |
| | Number of family meals to have per week |
| | Number of plants to care for in the home |
| | Number of new skills or hobbies to start learning each year |
| | Number of social events attended each month |
| | Number of health check-ups scheduled annually |
| | Number of meals cooked at home each week |
| | Number of times to change bed linens in a month |
| | Number of days per week dedicated to device-free time |
| | Percentage of clothing purchases that are from sustainable brands each year |
| | Number of cups of water to drink in a day |
| | Number of personal emails to send in a week |
| | Number of hours to listen to music in a day |
| | Number of journal entries to write in a month. |

| Category | Concepts |
|---|---|
| **Wealth and Economic habits** | Dollars of tax evaded by a person in a year |
| | Number of credit cards owned by a person |
| | Percentage of income saved annually |
| | Number of times a person shops online in a month |
| | Amount of money spent on dining out in a month |
| | Number of times a person checks their bank account balance in a week |
| | Number of loans taken out in a lifetime |
| | Dollars spent on impulse purchases in a month |
| | Dollars spent for buying electronics in an year |
| | Percentage of salary spent on housing |
| | Dollars of total saving in a year |
| | Number of luxury items purchased in a year |
| | Amount of money donated to charity annually |
| | Number of times a person reviews their budget in a month |
| | Percentage of income spent on entertainment |
| | Number of times a person consults a financial advisor in a year |
| | Amount of debt carried by a person on average |
| | Number of times a person uses a coupon in a month |
| | Amount of emergency savings recommended for a person |
| | Number of investment accounts owned |
| | Percentage of income spent on travel annually |
| | Number of times a person revises their will in a lifetime |
| | Number of financial seminars or workshops attended in a year |
| | Amount of money spent on subscriptions in a month |
| | Number of times a person renegotiates their salary in a career |
| | Number of times a person invests in stocks in a month. |

| Category | Concepts |
|---|---|
| **Environmental Sustainability** | Number of trees planted by a person in a year |
| | Number of times a person uses a reusable shopping bag in a month |
| | Amount of water saved by using water-efficient fixtures in a year |
| | Number of days a person participates in carpooling in a month |
| | Amount of energy saved by using energy-efficient appliances in a year |
| | Number of plastic bottles recycled by a person in a month |
| | Percentage of household waste composted |
| | Number of times a person rides a bicycle instead of driving in a week |
| | Amount of food waste reduced by a person in a month |
| | Number of times a person participates in community clean-up events in a year |
| | Percentage of products purchased that are made from recycled materials |
| | Number of times a person uses public transportation in a week |
| | Amount of greenhouse gas emissions reduced by using renewable energy sources in a year |
| | Percentage of clothing purchased that is second-hand or sustainably made |
| | Number of times a person participates in environmental advocacy or activism in a year |
| | Number of times a person chooses eco-friendly packaging options in a month |
| | Percentage of cleaning products used that are eco-friendly |
| | Number of times a person opts for plant-based meals in a week |
| | Amount of money spent on supporting environmental causes in a year |
| | Number of times a person uses single-use plastic in a week |
| | Amount of food waste thrown away in a month |
| | Number of times a person leaves lights on in empty rooms in a day |
| | Number of disposable coffee cups used in a month |
| | Amount of water wasted by leaving taps running in a month |
| | Amount of fuel wasted by idling a car in a week |
| | Number of times a person fails to separate recyclables from regular trash in a month. |

| Category | Concepts |
|---|---|
| **Politics and international relationships** | Number of international conflicts in a year |
| | Number of treaties or agreements signed by a country in a year |
| | Number of times a person votes in national elections in a lifetime |
| | Number of diplomatic visits made by a country's leaders in a year |
| | Percentage of a country's budget allocated to defense spending |
| | Number of international organizations a country is a member of |
| | Number of international trade agreements signed in a year |
| | Percentage of foreign aid given by a country as a portion of GDP |
| | Number of times a person participates in political protests in a year |
| | Number of bilateral meetings held between countries in a year |
| | Number of sanctions imposed by a country in a year |
| | Percentage of citizens who support international cooperation |
| | Number of diplomatic embassies a country maintains worldwide |
| | Number of refugees accepted by a country in a year |
| | Number of international espionage incidents reported in a year |
| | Number of military bases a country has abroad |
| | Percentage of international agreements ratified by a country's parliament |
| | Number of international cultural exchange programs sponsored in a year |
| | Number of cyberattacks attributed to foreign governments in a year |
| | Number of international humanitarian missions a country participates in a year |
| | Number of trade disputes resolved through international arbitration in a year |
| | Number of international human rights organizations criticizing a country's policies in a year |
| | Number of times a country is accused of violating international law in a year |
| | Number of military conflicts a country initiates in a year |
| | Number of times a country faces international boycotts due to its policies in a year |
| | Percentage of the population living under undemocratic regimes. |

| Category | Concepts |
|---|---|
| **Technology and Innovation** | Number of smartphone models that sold more than 10,000 pieces in a year |
| | Average number of hours people spend on social media per day |
| | Number of new technology products introduced to the market in a year |
| | Average age at which people purchase their first smartphone |
| | Percentage of households with smart home devices |
| | Average number of apps installed on a smartphone |
| | Number of electric vehicles sold in a country in a year |
| | Average number of hours people spend on online gaming per week |
| | Percentage of households with high-speed internet access |
| | Number of people using wearable fitness trackers in a country |
| | Average lifespan of a smartphone before being replaced |
| | Percentage of people using online banking services |
| | Number of streaming service subscriptions per household |
| | Average number of data breaches affecting consumers per year |
| | Percentage of consumers using mobile payment systems |
| | Average number of times people upgrade their tech devices in a year |
| | Number of people using telemedicine services in a country per year |
| | Percentage of market share held by electric vehicles |
| | Average amount of money spent by consumers on new technology annually |
| | Number of electric vehicle charging stations installed in a country per year |
| | Average number of hours people spend on virtual reality per week |
| | Percentage of consumers purchasing technology products online |
| | Number of broadband internet subscribers in a country |
| | Average number of new apps downloaded per person per year |
| | Number of households using renewable energy technology. |
| **Pet Care and Ownership** | Number of animals rescued and adopted in a year |
| | Average number of pets owned per household |
| | Amount of money spent on pet food annually |
| | Number of veterinary visits per pet per year |
| | Percentage of households with at least one pet |
| | Number of pet grooming sessions per year |
| | Amount of money spent on pet healthcare annually |
| | Number of pet-related products purchased per month |
| | Percentage of pets that are spayed or neutered |
| | Average lifespan of different pet species |
| | Number of times a pet is walked per day |
| | Amount of money spent on pet toys annually |
| | Number of pet-friendly parks or areas in a city |
| | Percentage of pets with microchips |
| | Number of pet training sessions attended per year |
| | Amount of money spent on pet insurance annually |
| | Number of pets abandoned or surrendered per year |
| | Percentage of pet owners who travel with their pets |
| | Number of pet-related accidents or injuries per year |
| | Average cost of pet adoption fees |
| | Percentage of households with multiple pets |
| | Number of pet-related events or expos attended per year |
| | Amount of money spent on pet boarding or daycare annually |
| | Number of pet adoptions from shelters versus breeders |
| | Percentage of pet owners who feed their pets homemade food. |

| Category | Concepts |
|---|---|
| **Travel, Tourism and Hospitality** | Number of countries visited by a person in their lifetime |
| | Average number of vacations taken per year |
| | Percentage of vacations that are international trips |
| | Number of cultural or heritage sites visited per year |
| | Average amount of money spent on travel annually in dollars |
| | Number of luxury cruises taken in a lifetime |
| | Percentage of travel done for leisure versus business |
| | Number of times a person stays at eco-friendly accommodations per year |
| | Average duration of an international trip in days |
| | Number of languages a person learns basic phrases of for travel |
| | Number of travel blogs or reviews written by a person in a lifetime |
| | Number of adventure or extreme sports tried while traveling |
| | Average number of travel souvenirs collected per trip |
| | Percentage of travel plans made spontaneously versus planned in advance |
| | Number of times a person travels with family per year |
| | Number of times a person visits the same destination multiple times |
| | Number of travel cancellations or delays experienced in a year |
| | Amount of money lost due to travel scams or fraud in a lifetime |
| | Number of times a person experiences food poisoning while traveling |
| | Number of travel insurance claims filed in a year |
| | Percentage of vacations that end with dissatisfaction or complaints |
| | Number of countries visited where a person experiences significant cultural differences |
| | Number of travel destinations visited due to trending social media recommendations |
| | Number of times a person misses a flight or train in a lifetime |
| | Amount of money spent on unexpected travel expenses annually |
| | Number of positive travel reviews written in a year. |

