# OpenReview forum: "Theory of LLM sampling: part descriptive and part prescriptive"
_ICLR.cc/2025/Conference — ICLR 2025 Conference Withdrawn Submission_

### Official Review · Reviewer_h6DC · 2024-10-27

**Soundness:** 3
**Presentation:** 3
**Contribution:** 2
**Rating:** 3
**Confidence:** 4

**Summary:**

The authors propose that we unpack LLM output into a descriptive and prescriptive norm. While the descriptive norm is the usual focus, the paper develops the notion of an parametric "ideal" that also influences output. They frame this as "sampling" from a set of possible outputs and connect the notions of the true distribution, the ideal distribution, and the sampling distribution. This theory may explain LLM output in three experimental settings (novel concepts, existing concepts, and prescriptive components) across 10 domains with 15 LLMs, in which the LLM provides average values, ideal values, and sample values. The sample values tend to be lie between average and ideal values. This is replicated in human subjects experiments, though there is limited correlation between the contexts in which humans versus LLMs have more tendency towards idealized values.

**Strengths:**

1. The authors provide robust evidence for their central claim that LLMs tend towards rather than away from idealized numbers when providing samples. There is plenty of surface area to critique, but no single paper could address every concern for a claim like this.
2. The presentation is clear. I appreciated the color-coding in particular (it may not be color-blind accessible, but I don't see how that could be fixed).
3. I appreciate all the detail in the appendix.

**Weaknesses:**

1. The scope of the contribution is marginal. It is unsurprising that LLMs tend towards idealized output (e.g., arxiv.org/abs/2406.17055), and I'm skeptical that this contribution, no matter how well-evidenced, would be of sufficient value and significance for ICLR. A more compelling contribution in this vicinity, just as an example, would be the topic of Figure 4 and Figure 5. I think there is too much noise in the examples (e.g., categories in Table 2) to conclude that there are systematic differences, but I don't know in what contexts LLMs would produce examples more idealized than those humans produce, and this could have interesting implications (e.g., that preference-tuning has prioritized some moral issues more than others).

2. The prototypicality experiment seems out of step with the others and particularly weak. The authors themselves accept that the idea of prototypicality involves both the descriptive and normative factors, so why wouldn't the model agree with that? It looks like the prompts for prototypicality are "good," "paradigmatic," and "prototypical," which are clearly all words that tend towards the normative.

3. While I'm convinced of the central claim of the paper, I'm not sold on many of the methodological details. For example, I don't find the paragraph Lines 247-254 convincing. Why would an estimate of zero hours per week "glubbing" mean "no pre-existing statistical association"? Doesn't the model presumably recognize that it is not a real hobby, and if it were (e.g., a recent social trend past its training data), it would most likely not be common enough to budge the average over 0? Moreover, wouldn't most hobbies except perhaps the most popular (e.g., running) have mean 0? This is not to mention that the prompt, "What is the number of hours a person does glubbing in a week?" is ambiguous (e.g., average person? a person who does glubbing?). The "C+" assessment is similarly unconvincing; why is it assumed that I see that as making "glubbing" neutral? One better direction would be using a variety of made-up words, ideally with different structure (e.g., some being more realistic like "tally ball"; some being assigned only a code like "Hobby 9235").

4. I don’t find the human subjects studies very convincing. This may just be missing methodological details (e.g., how did the authors ensure the human subjects were providing truthful responses? This could be methods such as Bayesian truth serum (science.org/doi/10.1126/science.1102081)), but if I’m understanding correctly, the average human respondent said that, ideally, 11.1% of people would drive drunk, which is missing face validity. I also did not see a discussion of the institutional ethics process for the human subjects studies.

5. I was not sold on the motivation for this "sampling" process in the first place. As I said, I'm not skeptical of the central claim, but why would we care how the model responds to, “What is the first number of glubbing hours that comes to your mind”? That's not a realistic use case. Something in this vicinity that would be more interesting to study would be the generation of user personas, which is common (e.g., arxiv.org/abs/2310.11667) and could be probed in similar ways.

**Minor weaknesses**

1. The presentation of the paper is limited by excessive mathematical notation when everyday language would suffice. $C$ could be removed from most places and just talk about the concept in plain English (or something like capitalized terms if you prefer). There are little oddities like saying "Gaussian" when "Normal" would suffice or saying "we use a binomial test" for a single sample. Equations (1) and (2) seem unnecessary and make it seem more complicated than it is. This reads like a paper trying to seem more mathematical than it is, and I'd encourage the authors to just embrace having a less technical contribution.

2. Similarly, I found the "concept" versus "category" distinction confusing.

3. It's not clear to me how variation was created across "samples." Was that purely temperature variation, or something like prompt variation? I don't think the log-probs assigned to various tokens should be expected to be a reliable sample of something like the real-world noise in human responses.

4. The use of "ideal prompt" in the appendix, which seems to just mean "main prompt," is confusing given the use of "ideal" elsewhere.

Typos:

“$ cheated on taxes”

“Findings in prior art”

"thought the ...  converge" -> "though the ... converge,"

**Questions:**

See Weaknesses.

---

> ### Author Response · Authors · 2024-11-21
>
> > The scope of the contribution is marginal.  It is unsurprising..
>
> On the contrary, we believe the scope of the contribution is anything but marginal. We view LLMs as a new form of intelligence (Embers of Autoregression: Understanding Large Language Models, reference: https://arxiv.org/abs/2309.13638,) and use grounded methodologies and principles from cognitive science to derive a theory which explains the heuristics behind LLM sampling its options. The topic of our study is inspired from the experiments done in possibility sampling in humans (refer: How we know what not to think? https://www.cell.com/trends/cognitive-sciences/fulltext/S1364-6613(19)30231-1,
> What comes to mind? https://www.sciencedirect.com/science/article/abs/pii/S0010027719302306) where it was investigated how indeed do humans narrow down options from the vast set of possibilities, most of which go unrealised. For example: If a car breaks down in the middle of the road, an agent faces innumerable decision-making options. It could either call for roadside assistance or a tow truck service on one end or attempt to push the car off the road and walk to the destination at the other extreme. It is difficult to consider/deliberate on all possible options. To narrow these options to a `consideration set’ within a computationally reasonable time, the agent needs to employ a heuristic driven approach (As discussed in the cognitive science papers referenced above).
>
> We investigate this thoroughly in LLMs and conclude that the heuristics used by LLM converges with that of humans. We observe that, both these decision making systems use a heuristic of both descriptive and prescriptive norms. We believe this convergence (and divergence in the exact prescriptive norms) uncovers the nature of decision making of LLMs, as a similar framework was used to explain the heuristics behind decision making by humans.
> Furthermore, the proposed theory has immediate practical implications like the medical case study presented in the paper. Besides multiple similar use cases, the theory would affect the next wave of synthetic data generation.This strengthens the claim that the paper has actionable insights.

---

> > ### Author Response · Authors · 2024-11-21
> >
> > > The contribution is different from the arXiv paper : arxiv.org/abs/2406.17055
> >
> > Moreover, the other paper that you shared doesnt share the same theory. Our paper argues for LLMs seeming to use the heuristic of descriptive and prescriptive norm in its outputs whereas the other task is of rationality and expected value computation using weighted probability and value explicitly given to the LLM. That’s different from what we are proposing in the paper where we say System1 cognition of an LLM when sampling its options from innumerable ones from its weights has descriptive and prescriptive components.
> >
> > >The authors themselves accept that the idea of prototypicality involves both the descriptive and normative factors,
> >
> > Please refer to the original prototypical paper by Rosch, Eleanor, and Carolyn B. Mervis (1975) Family Resemblances: Studies in the Internal Structure of Categories." Cognitive Psychology, 7(4), 573–605 and Barsalou, Lawrence W. (1985), "Ideals, Central Tendency, and Frequency of Instantiation as Determinants of Graded Structure in Categories. Both of these **do not say** prototypes have a prescriptive component, so the words themselves **do not carry** a normative meaning. This is a recent finding by Bear et al Normality: Part descriptive, part prescriptive that the **concept of prototypicality** has a prescriptive component and we have used the same methodology to see how LLMs represent their concepts and they too have the same structure in their representations imitating humans. Also Bear et al perfromas the experiment on humans. Our experiment is still important as all findings on human heuristics are not to be assumed as shared by LLMs.
> >
> > >  Why would an estimate of zero hours per week "glubbing" mean "no pre-existing statistical association"?
> >
> > We associate the consistent reporting of zero on entirely fictional concept with the LLM having no associated data; as the model having no pre-existing distribution, statistical trend, or observed examples to reference. The LLM, due to its pre training in statistical modeling theories might have some notion of null hypothesis. The concept of zero is often used as a representation of a null hypothesis—meaning there is no effect or no established relationship. In the case of new concept, since there is no background data, the LLM essentially adopts a null hypothesis that "there is no engagement in glubbing." The response of zero thus might correspond to the absence of evidence or statistical trend to suggest otherwise. We understand the concern you might have and will remove this claim from the main draft. This does not affect the experiment as the observations still confirm the theory.
> >
> > We’ve tried 100 other made up words in the prompt robustness section of the paper and they return similar results. **We have also tried with no token at all** (without specifying the concept name), giving just grades along distribution and this sample too follows the same heuristics thereby proving the phenomenon does not rely on specific tokens of prompting. The observed phenomenon is not a product of this choice of word or the concept (already in appendix) or the structure of the prompt itself (already in appendix).

---

> > > ### Author Response · Authors · 2024-11-21
> > >
> > > > I don’t find the human subjects studies very convincing.
> > >
> > > The data of humans studies are taken from an already peer reviewed study and we believe they followed the necessary ethical precautions needed for the study. This is stated with reference in the text. Please note that the experimental setup with prompts are scaled up from the same papers.
> > > > I was not sold on the motivation for this "sampling" process in the first place. Why would we care how the model …
> > >
> > > As mentioned earlier, the **sampling experiment is designed to be controlled and focused on eliminating real world concepts which have implicit associations**. The particular prompt that you mentioned is a direct replication of the prompt from the original paper but we sample from a number of prompts so as to provide robustness. We **arent proposing a new methodology, rather deriving these grounded experiments from widely accepted methodologies from a number of papers** in the field of cognitive science. (Please refer Normality: Part descriptive, part prescriptive A Bear, J Knobe, How We Know What Not To Think
> > > Jonathan Phillips Adam Morris  Fiery Cushman,What comes to mind? Adam Bear, Samantha Bensinger, Julian Jara-Ettinger, Joshua Knobe, Fiery Cushman). By scaling up this experiment and doing this more rigorously in LLMs, we infer the same inference done for humans in the original study that LLMs seem to mimic humans in having a descriptive and a prescriptive component. This critical experiment is not to show the practical applications of the theory (which is done in following experiments and case study).
> > >
> > > > Question: There are little oddities like saying "Gaussian" when "Normal".
> > >
> > > We clarify this in the paper. We have tried to reduce the complexity of terms but keep the word gaussian as the word normal might be confused with the notion of normal in normative.
> > > > Question 2
> > >
> > > We change the usage of the word concept and category to have more consistency. Thank you for pointing it out.
> > > > Question 3: Creating variation in samples.
> > >
> > > We conduct the experiment with temperature zero and 0.8 as reported in the paper. The variation in samples is caused by the distribution being different. We sample from a gaussian of the same mean and std in each run.
> > > > Question 4
> > >
> > > We are not sure if we understood the question. We do not use ‘main prompt’ in the paper. Ideal is used to invoke the idea of prescriptive norms, as in prior art.

---

> > > > ### Comment · Reviewer_h6DC · 2024-11-21
> > > > **Reply to authors**
> > > >
> > > > Please see my other comment.
> > > >
> > > > > The data of humans studies are taken from an already peer reviewed study...
> > > >
> > > > Data is not reliable merely because it was peer-reviewed.

---

> > > > > ### Author Response · Authors · 2024-11-22
> > > > > **Replying to : Data is not reliable merely because it was peer-reviewed**
> > > > >
> > > > > Bear et al. has also given the raw data for the experiment. We have cross checked these results with the raw data and the values reported in the paper seems correct.

---

> > > ### Comment · Reviewer_h6DC · 2024-11-21
> > > **Reply to authors**
> > >
> > > Please see my other comment.
> > >
> > > > Please refer to the original prototypical paper...
> > >
> > > If your argument were that some cognitive psychologists see prototypicality as exclusively descriptive, and this is incorrect, that's a paper that should be written about humans focused on human data, not LLMs. Moreover, if this is your reading of these works, it needs much better justification. E.g., this is not consistent, "It also serves as a mental benchmark, embodying both statistical regularities and goal-oriented ideals within a category (Barsalou, 1985)." You may see "goal-oriented ideals" as non-normative, but that would be an unusual take that I'm not sure could be justified.
> > >
> > > > We associate the consistent reporting of zero on entirely fictional concept with the LLM having no associated data; as the model having no pre-existing distribution, statistical trend, or observed examples to reference...
> > >
> > > I don't see a justification here that the distribution with all probability mass at zero is a lack of a pre-existing statistical association. That is a very particular statistical association.

---

> > ### Comment · Reviewer_h6DC · 2024-11-21
> > **Reply to authors**
> >
> > I have read the authors' rebuttal and will maintain my score. I don't see in this rebuttal any arguments or data that were not already in the paper, so I see no reason to change my view.

---

> ### Author Response · Authors · 2024-11-21
> **Clarifying the discussion on the prescriptive component for prototypes**
>
> Thank you for the reply and for giving us a chance for further clarification. We believe we can add this discussion to the appendix to clarify for future readers interested in this nuance.
> But before going into the details we would like to answer your two questions:
>
> - “Prototypicality involves both the descriptive and normative factors, so why wouldn't the model agree with that?”
> It is not obvious that representation of concepts in the LLM exactly follows humans. The notion of ideality in prototypes in humans is not coming from the language token but the representation of the concept by humans (like the example: for humans robin is a prototype bird).
> - Secondly, we would like to clarify that our argument is not that some cognitive psychologists see prototypicality as exclusively descriptive. But we were trying to clarify that the idea is more nuanced. We understand the confusion and describe this in more detail. Please understand that in the process of describing this we are not making psychology claims but clarifying this idea:
>
>
> The perspective we want to convey from Barselou: we directly quote two instances from the paper. "Whereas ideals may determine a category's graded structure in one context, central tendency may determine a different graded structure in another", and “So far, no experiments have shown that ideals are causal determinants of graded structure".  This means according to Barsalou, when sampling, humans wouldn't use both prescriptive and descriptive prototypical ratings in the same context. But, Bear et al (Normality: Part descriptive, part prescriptive) show that human concepts have both components in the same context in a unified representation, providing an insight into how humans think about concepts, and our notion of normality is in fact both prescriptive and descriptive. When we try to rate a normal teacher, we include both prescriptive and descriptive components in the same context.
>
> Given the two different theories, we test this theory in LLMs. Previous experiments in this paper show that LLMs, when sampling from innumerable options, use both prescriptive and descriptive norms as a heuristic in the same context (Figure 1) akin to a unified representation. We show similar results of how prototypicality rating also has the same unified representation of both prescriptive and descriptive norms in the same context.

---

> ### Author Response · Authors · 2024-11-22
> **On pre-existing statistical association**
>
> We appreciate the reviewer’s critique regarding the interpretation of the consistent reporting of zero for fictional concepts. It is difficult to evaluate pre-existing statistical associations, we recognize that the model's ability to identify the fictional nature of terms like 'glubbing' may sufficiently explain the lack of implicit associations. Specifically, the language model likely recognizes that 'glubbing' does not correspond to any real-world concept in its training distribution, which naturally results in outputs that reflect the absence of relevant statistical patterns. Further comparison of the effect and control experiment (in experiment 1)  provides the evidence for the theorem when using these new concepts.

---

> ### Author Response · Authors · 2024-11-27
> **Incorporating changes and addressing concerns**
>
> We thank you for sharing your initial concerns and for engaging in a constructive discussion. In response to your concerns, we have also made substantial changes to the paper:
> We believe we had posted clarification on the five concerns you listed
> - Implication of the work: We have expanded the introduction and related work sections to better articulate the broader implications and contributions of our study, also in the discussion in rebuttal as acknowledged by sdiR
> - We had a constructive discussion on the motivation for the prototypicality experiment also included in the paper.
> - We readjusted the validation of pre-existing statistical associations. We added this to the discussion and specifically to L255.
>  - To the best of our knowledge, presuming trust in the peer reviewed scientific work is common practice. A general mistrust in empirical evidence would undermine scientific progress.
> - We have clarified that the critical experiment is meant to empirically show the effect of proposed theory isolating confounders. We add realistic questions like `Pick a sample to represent the given distribution’ and ten other phrasings and a case study to show the applications.
>
> If all the concerns are addressed we kindly request you to reconsider the rating or engage in further discussions.

---

> > ### Comment · Reviewer_h6DC · 2024-11-27
> >
> > > To the best of our knowledge, presuming trust in the peer reviewed scientific work is common practice. A general mistrust in empirical evidence would undermine scientific progress.
> >
> > I just want to be particularly clear on this point, that I do not see this phrasing as an accurate representation of my concern. There is a wide spectrum of reliance we can have in published research. Clearly we should not completely mistrust or trust it. The human subjects data, which I had originally taken as a contribution of the current work (and I would encourage the authors to make the wording around 462-465 clearer that this is not the case), is a significant part of the argument presented by the authors, and that does mean it needs to pass muster in order for the current work to be considered a sound and significant contribution.
> >
> > To engage briefly with the "undermine scientific progress" framing, if we were to take all peer-reviewed work as reliable, that would in fact undermine scientific progress. Entire streams of work could be built on that data without proper scrutiny. With the wide range of peer review standards used today, clearly we need to be quite judicious. Just because other people err does not mean we should.
> >
> > As I said, there is reason for mistrust when a study finds that people see 11.1% as the "ideal" number of people who drive drunk--which, to my knowledge, is widely considered an egregious and harmful act. I was not able to clarify this issue by looking briefly at Bear et al. (2020), as their Table 1 that looks very similar to the current work's Table 4 does not have the same numbers. (I am charitably assuming that there is a legitimate reason for the tables having different numbers, and the current work not merely reproducing the Bear et al. (2020) table that has the same rows and columns but different numbers.)

---

> > > ### Author Response · Authors · 2024-11-27
> > >
> > > Thank you so much for your due diligence and pointing out the discrepancy in  numbers between Table 4 in our appendix and that of Bear et al.
> > > We acknowledge that there is a mistake in the table, you are correct and we highly appreciate it. As you rightly hypothesised the mean ideal drinking hours for a person is less (1.23 hours).
> > > When writing the appendix, we made a mistake in filling in this table with our raw data from different settings. We now updated the paper with the exact numbers. We genuinely apologize for our earlier message that we wrongly interpreted as mistrusting prior art and we missed this specific (completely valid) reasoning behind your concerns.
> > >
> > > However, please note all the plots (figure 4 in the main paper) use the correct values (which you can verify by the attached notebook https://anonymous.4open.science/r/LLM_Norm-E070/Analyse.ipynb. We also upload our entire code fro this experiment, which we will make available for reproducibility, if you would like to verify our setup and numbers: https://anonymous.4open.science/r/LLM_Norm-E070/. We now also verified our other results. Please also note that we just originally aimed to copy the table for convenience (we again sincerely apologize for the mistake on our part). We add a disclaimer in the caption that these results are from Bear et al.
> > >
> > > Please also note we didn’t make claims about the conclusions of Bear at al. that were not stated by the authors. Our foundation is based on the findings clearly expressed by the authors, not our interpretation of their results (which we used correctly in our experiments).
> > >
> > > While we acknowledge this and are very thankful for your diligence, we would like to emphasize that this in no way changes the conclusions of our work and that of Bear et al. Our conclusions about the existence of biases in LLMs and the misalignment between humans and LLMs (based on figure 4) stay the same.
> > >
> > > Thank you again for your constructive questions and discussion, and please check our updated paper.

---

> > > > ### Comment · Reviewer_h6DC · 2024-11-27
> > > >
> > > > Thanks for looking into it. If I may give friendly advice for your next submissions, I think one's first reaction to a reviewer pointing out a questionable figure should be to diagnose the source of the issue and make corrections, rather than to jump to a broad appeal that is beyond the scope of the work, like the reliability of peer review.
> > > >
> > > > In general, I still do not see compelling reasons to change my score, but I wish the authors the best for this and future work.

---

> ### Author Response · Authors · 2024-12-01
>
> Thank you for pointing out the error in quoting the results from prior art in an Appendix table. We hope the new corrected version of the paper uploaded clarifies the earlier concern on the validity of human studies. We request the reviewer to bring up the concerns for not improving the score.
> To clarify our understanding of the concerns you’ve  raised initially:
> - We’ve pointed out why the current work is markedly different from an arxiv paper you had mentioned, our paper talks about uncovering heuristics while the other is aimed at the assumptions llms have about humans.
> -Implication of the work: We have expanded the introduction and related work sections to better articulate the broader implications and contributions of our study, also in the discussion in rebuttal as acknowledged by sdiR. Please review the updated version.
> - We had a constructive discussion on the motivation for the prototypicality experiment also included in the paper.
> - We readjusted the validation of pre-existing statistical associations. We added this to the discussion and specifically to L255.
> - The current version amends the error in quoting prior art. The results in the main text and the conclusion remain the same allaying your concerns regarding the human experiments.
> - We have clarified that the critical experiment is meant to empirically show the effect of proposed theory isolating confounders. We add realistic questions like `Pick a sample to represent the given distribution’ and ten other phrasings proving robustness and a case study to show the applications.
>
> We would like to hear your feedback for the lower score.
>
>
> We would like to emphasis on the implications of the work. We view LLMs as a new form of intelligence (Embers of Autoregression: Understanding Large Language Models, reference: https://arxiv.org/abs/2309.13638,) and use grounded methodologies and principles from cognitive science to derive a theory which explains the heuristics behind LLM sampling its options. The topic of our study is inspired from the experiments done in possibility sampling in humans (refer: How we know what not to think? https://www.cell.com/trends/cognitive-sciences/fulltext/S1364-6613(19)30231-1, What comes to mind? https://www.sciencedirect.com/science/article/abs/pii/S0010027719302306) where it was investigated how indeed do humans narrow down options from the vast set of possibilities, most of which go unrealised. For example: If a car breaks down in the middle of the road, an agent faces innumerable decision-making options. It could either call for roadside assistance or a tow truck service on one end or attempt to push the car off the road and walk to the destination at the other extreme. It is difficult to consider/deliberate on all possible options. To narrow these options to a `consideration set’ within a computationally reasonable time, the agent needs to employ a heuristic driven approach (As discussed in the cognitive science papers referenced above). We investigate this thoroughly in LLMs and conclude that the heuristics used by LLM converges with that of humans. We observe that, both these decision making systems use a heuristic of both descriptive and prescriptive norms. We believe this convergence (and divergence in the exact prescriptive norms) uncovers the nature of decision making of LLMs, as a similar framework was used to explain the heuristics behind decision making by humans. Evidence that LLMs process information in ways resembling human heuristic reasoning has profound implications. Furthermore, the proposed theory has immediate practical implications like the medical case study presented in the paper. Besides multiple similar use cases, the theory would affect the next wave of synthetic data generation. This strengthens the claim that the paper has actionable insights.
>
> We would like to bring to your notice that we have updated our draft following the reviews and comments. With the extended discussion period, we are glad to answer further questions. We again thank the reviewer for the constructive feedback. If our clarifications address your concerns, we request you to consider improving the score.

---

### Official Review · Reviewer_szCF · 2024-11-04

**Soundness:** 2
**Presentation:** 2
**Contribution:** 2
**Rating:** 5
**Confidence:** 4

**Summary:**

This paper investigates the sampling mechanisms of Large Language Models (LLMs) and introduces a theory suggesting that, like humans, LLMs generate responses using a combination of descriptive norms (reflecting statistical averages) and prescriptive norms (representing idealized values). The authors conducted a series of experiments to validate this theory.

**Strengths:**

The problem they address, the numerical sampling behavior of LLMs and the biases that influence it, is under-explored and interesting. Framing this problem within cognitive science offers a new perspective. The observations about LLM’s sampling abilities are somewhat novel.

**Weaknesses:**

I believe the first half of the paper (before the experiments) could be more clearly presented with some editing. The abstract talks about heuristics in LLMs’ sampling, but it was not clear what ‘sampling’ refers to--for example, whether it pertains to token generation or other outputs. Additionally, while the authors repeatedly mention the prescriptive component and the notion of an ideal, they don’t provide concrete examples or sufficient details to clarify these concepts. As a result, readers must wait until the experimental sections to fully understand key ideas presented in Sections 1 and 3. I suggest including an example at the beginning to motivate the problem better and increase the clarity and engagement.

The notations used in the paper, such as the various forms of 'C,' can be confusing and require readers to frequently refer back to their definitions, which disrupts the flow of reading.

Although their observations are interesting, the implications of this study are unclear. How can we further investigate the internal mechanisms of the LLMs based on these results?

**Questions:**

What happens if we change the scale of sampling numbers? For example, if we change the scale of numbers from 1-100 to something else.

How do the distributions of the numbers that LLMs sampled look like? Providing a more detailed analysis of the distributions of the sampled numbers would be valuable. Do LLMs exhibit a tendency to generate certain numbers more frequently? How does this behavior vary across different models or parameters, such as temperature settings?

Regarding the explanation in line 300 about assigning random grades, could you clarify what this entails? Does it mean that the grades are assigned without any meaningful relationship to the numbers?

---

> ### Author Response · Authors · 2024-11-21
>
> > Lack of clarity before experiment section
>
> We sincerely thank the reviewer for their valuable feedback, particularly highlighting areas where clarity and engagement in the paper can be improved. Following the comment, we redo our introduction section and restructure the sections to improve the engagement.
> Following reviewer suggestion we add a paragraph for giving concrete definitions in the main text and establish all the ideas used in the paper along.  We define sampling, descriptive/ideal and prescriptive/average. Later we also define prototypes later on in the respective section. We clarify the notation of ‘C’ upfront to clarify the usage. If you meant the use of C_a for average and so on, we will change this to A_c for camera ready. Currently changing it might cause confusion for reviewers. We believe the changes have helped improve the readability and engagement of the paper.
> Here are the definitions.
> We propose a theory that the sampling of an LLM is driven by a descriptive norm  (the notion of statistical average) and a prescriptive norm (a notion of an ideal represented in the LLM)(Figure teaser).
> We define response sampling as the process by which the model probabilistically selects outputs from a distribution of potential responses.
> A descriptive norm represents what is commonly observed or statistically likely within a given context, reflecting the frequency or probability of events or behaviors without implying any value judgment.
> A  prescriptive norm represents an implicit standard of what is considered ideal, desirable, or valued within a context, often encoded by grades/scores that prioritizes outcomes deemed "better/optimal".
> The proposed theory implies, the sample of an LLM not only reflects the statistical regularities of the data descriptive norms) but also systematically incorporates an idealized version of the concept prescriptive norms).
>
> >Implications of the study
>
> We thank the reviewer for their feedback. We believe the theoretical foundation established in this paper is crucial before addressing empirical implications or adaptations. Here, the critical experiment (experiment -1) of sampling using numbers without any implicit associations is inspired from established principles from prior art in cognitive science to show LLMs use prescriptive norms as a heuristic when sampling their outputs. Apart from the direct case study of medical recovery times of patients being undersampled , we believe this has implications for synthetic data generation when LLMs consistently use prescriptive norms when they sample data which the user might be misled and implicitly think of as a descriptive average.
> We believe our work opens the doors for investigating the effects of prescriptive norms in various applied tasks. The purpose of this work was to establish the theoretical foundations of this heuristic being present. That said, since we are hypothesizing the existence of a prescriptive component we believe future work in mechanistic interpretability can steer a possible “prescriptive vector” when the output might be biased regarding a particular instance.  We believe our approach offers a similarly useful starting point for studying this aspect of LLMs and hope to see future work that goes deeper into this mechanism.

---

> ### Author Response · Authors · 2024-11-21
>
> > Interpretation of the study
>
> We also view LLMs as a new form of intelligence (Embers of Autoregression: Understanding Large Language Models, reference: https://arxiv.org/abs/2309.13638,) and use grounded methodologies and principles from cognitive science to derive a theory which explains the heuristics behind LLM sampling its options. The topic of our study is inspired from the experiments done in possibility sampling in humans (refer: How we know what not to think? https://www.cell.com/trends/cognitive-sciences/fulltext/S1364-6613(19)30231-1,
> What comes to mind? https://www.sciencedirect.com/science/article/abs/pii/S0010027719302306) where it was investigated how indeed do humans narrow down options from the vast set of possibilities, most of which go unrealised. For example: If a car breaks down in the middle of the road, an agent faces innumerable decision-making options. It could either call for roadside assistance or a tow truck service on one end or attempt to push the car off the road and walk to the destination at the other extreme. It is difficult to consider/deliberate on all possible options. To narrow these options to a `consideration set’ within a computationally reasonable time, the agent needs to employ a heuristic driven approach (As discussed in the cognitive science papers referenced above).
> We investigate this thoroughly in LLMs and conclude that the heuristics used by LLM converges with that of humans. We observe that, both these decision making systems use a heuristic of both descriptive and prescriptive norms. We believe this convergence (and divergence in the exact prescriptive norms) uncovers the nature of decision making of LLMs, as a similar framework was used to explain the heuristics behind decision making by humans. Evidence that LLMs process information in ways resembling human heuristic reasoning we believe is not just a fun fact.
> Furthermore, the proposed theory has immediate practical implications like the medical case study presented in the paper. Besides multiple similar use cases, the theory would affect the next wave of synthetic data generation. This strengthens the claim that the paper has actionable insights.
>
> > Questions: change the scale of sampling numbers (rounded for easier observation)
>
> | Range       | Average | Pos Ideal | Neg Ideal |
> |-------------|---------|-----------|-----------|
> | 1-100       | 45      | 46        | 31        |
> | 100-200     | 145     | 152       | 143       |
> | 200-300     | 245     | 261       | 241       |
> | 300-400     | 345     | 361       | 344       |
> | 400-500     | 445     | 489       | 442       |
> | 500-600     | 545     | 549       | 514       |
>
> > Distribution generated by LLM
>
> The distribution  generated by LLM is generally comparable to the input distribution except the higher standard deviation. But we do not claim this as a finding. We will add the original distributions and a discussion in the appendix. Here we give the standard deviation for input having std 15
>
> | Range       | Average | Std |
> |-------------|---------|-----|
> | 1-100       | 45      | 21  |
> | 100-200     | 145     | 24  |
> | 200-300     | 245     | 19  |
> | 300-400     | 345     | 20  |
> | 400-500     | 445     | 23  |
> | 500-600     | 545     | 23  |
>
> > Random grades for control experiment
>
> Yes, we assign random grades. We add more details in the revised draft. We have three control settings to validate the findings of the experiment.
>
> In the first control we assign random grades to values, and show that sampling is driven by probabilities alone.
> In the second control we give no grades with the numbers and again show that the sample is driven solely by the likelihood of the distribution.
> In the third control experiment, we assign the mean Cµ with the highest grade and lower grades for increasing distance from this value. This means the most likely value is also the most ideal value. (we observe sample has no bias, indeed the sample mean is the same as the input mean).
> As a sanity check we ask the LLM to report averages (to validate that the observation is not due to the inability to mathematically compute averages.)
> We again thank the reviewer for their detailed review and constructive feedback. If our clarifications address your concerns, we request you to consider improving the score.

---

> ### Author Response · Authors · 2024-11-27
> **Revised draft, addressing concerns.**
>
> Thanks again for the questions. We would like to bring to your notice that we have submitted the rebuttal and updated our draft following the reviews and comments. With the extended discussion period, we are glad to answer further questions. We again thank the reviewer for the constructive feedback. If our clarifications address your concerns, we request you to consider improving the score.

---

### Official Review · Reviewer_Xi8p · 2024-11-05

**Soundness:** 1
**Presentation:** 2
**Contribution:** 1
**Rating:** 3
**Confidence:** 3

**Summary:**

Summary: This paper attempts to distinguish between systems 1 and system 2 reasoning (from cognitive literature) within LLMs using a prompt-based approach. The authors argue that sampling using LLMs is driven by a descriptive component (that is a statistical average) and a prescriptive component (an “ideal”), which they loosely tie back to these two systems of reasoning.

**Strengths:**

This work asks the question of whether there are parallels between system 1 and system 2 reasoning in humans and in LLMs. The question is interesting, and contextualizing it within sampling is novel.

**Weaknesses:**

I have a three concerns with this paper: the coverage of related work framing the motivation is sparse;  the methods may not necessarily result in the outcome supporting the hypothesis of different systems of reasoning; finally, definitional specificity seems to be absent.

While the work attempts to ask with fundamental questions of how an output is produced, it does not engage with prior work related to sampling (e.g. inducing distributions using LLMs is an active space of literature) or how outputs are produced (the so-called stochastic parrot discussion). Perhaps more importantly, there were no clear mapping system 1 and 2 to the setting of LLMs. The related works setting up and motivating this approach are tied to popular cognitive science theories, along with cherry picked cases where some systems adopt learned models (e.g. classifiers or LLMs) as inputs into other systems; this does not equate to distinguishing between System 1 and System 2 reasoning in an LLM.

The methodology adopted focuses on responses to prompts as motivation for a rather large claim regarding how heuristics shape sampling processes. This has led to a scenario black box prompting and subsequent response evaluation, which is relatively limited particularly when making claims regarding how an output is produced. In many of the evaluations, the method is tied to prompting itself. For example, in 4.3 prompts such as, “To what extent do you think this is an average C?” are adopted as metrics, when aggregated, for internal consistency and adopted as the measurement itself. Without additional evidence (e.g. internal state), I'm not confident this result is supporting the authors' claims.

How the authors define "ideal", which is a core contribution of the work, is unclear in the paper; in the abstract, there is a mention that the ideal is represented in the LLM, but this is not mentioned specifically later in the paper and it becomes unclear how the concepts of average and ideal are differentiated.

Additional comments: Grammar error: "without" is included 2x in first sentence. Some of the formatting decisions are interesting / atypical for ML papers, including this venue, such as coloring words of interest (I would suggest the authors switch to italics or boldening instead to avoid issues with printing e.g. in black/white and for color blind readers). The first figure is also rather difficult to parse.

**Questions:**

How are the authors differentiating between prototype, ideal, and statistical average? There are a few cases where the language seems to be used interchangeably and putting this early in the paper would be helpful.

---

> ### Author Response · Authors · 2024-11-21
>
> Thanks for the review and the references. Below we address the three main concerns of the reviewer
>
> > Coverage of prior art
>
> We include a detailed related work and discussion on the stochastic parrot explanation of LLM outputs. We also add discussions on the different studies that explain the biases in the output of LLMs.
>
> > more importantly, there were no clear mapping system 1 and 2 to the setting of LLMs.
>
> We understand your review as (a) ‘we are not having related work establishing the system-1 mapping of LLM output’: we discuss this in L083-088. (b) ‘There being no literature establishing this’: LLMs are conceptually likened to system-1 by [1] and various peer reviewed prior art. This idea has even been taken ahead by  Yao et al. by adding a symbolic component to induce system-2 behavior.
>
>
> > On using black box prompting methodology and response evaluation to make the claim
>
> We would like to point out we aren't the first ones to adopt this method of using outputs to make inference. In fact, a whole bunch of peer reviewed papers in top tier ML conferences and leading journals use the same methodology of inferences of mechanism/biases/heuristics through prompt outputs. We list some of them below:
>
> (Nature Human Behavior) https://www.nature.com/articles/s41562-024-01882-z
> Inference of the process of theory of mind through a battery of psychologically inspired tests
>
> NeurIPS 2023 - In-Context Impersonation Reveals Large Language Models' Strengths and Biases
> Prompt outputs recover mechanisms of human-like developmental stages of exploration
>
> NeurIPS 2024 - Measuring Implicit Bias in Explicitly Unbiased Large Language Models
> Propose a `new prompt-based measure drawn from psychology’s long history of research’. Paper designs a prompt based evaluation to expose nuanced biases in proprietary LLMs that explains implicit decisions in LLMs.
>
> NeurIPS 2024 - Cognitive Bias in High-Stakes Decision-Making with LLMs
> `Inspired by prior research in psychology and cognitive science’, the paper designs a prompt based evaluation strategy for explaining LLM decisions. The paper infers the LLMs have anchoring effect, the outputs can be explained by Status Quo maintaining tendency and Group Attribution Bias.
>
> ICML 2024 - Do Language Models Exhibit the Same Cognitive Biases in Problem Solving as Human Learners?
> Study relation to cognitive biases known in children when solving arithmetic word problems. Follows the LLM in a three stage process involved in solving math word problems: mental model, arithmetic expression and answer. They use a template filling based on prompts to construct the model and infer similar cognitive behavior.
>
> ACL 2023 - Mind the Biases: Quantifying Cognitive Biases in Language Model Prompting
> Study related to uncovering and quantifying biases - availability bias and framing bias are mechanisms which are inferred from prompt output.
>
> Furthermore, in experiment 1, we carefully construct a critical experiment where the result shows evidence for the proposed theory. We show the robustness of the observation to different variations of the prompt to make sure that the results are explained by the proposed hypothesis.
>
> On a final note, we see LLM as a Simulacra of human behavior (Refer: Role play with large language model https://www.nature.com/articles/s41586-023-06647-8) where LLM is effectively roleplaying its behavior and has no underlying intentions or goals. Through our grounded cognitive science experiments methodology, we find consistent patterns of behavior of LLM having a notion of value (prescriptive norm) in its outputs. This effect is observed even after testing for multiple variants of the same prompt  indicating the robustness of the findings. The LLM seems to mimic human cognitive processes of descriptive and prescriptive norms, effectively role playing them like a simulacra.
>
> > Definitional specificity seems to be absent.
>
> Thank you for this comment. We add the definitions used in the paper earlier as the second para of the introduction. We define sampling, descriptive/ideal and prescriptive/average. Later we also define prototypes later on in the respective section.
>
> > Grammar
>
> Thanks for the comments. We hope to have fixed the grammatical errors in the revised version.
>
> We again thank the reviewer for the feedback.

---

> ### Author Response · Authors · 2024-11-27
>
> Thanks again for the questions. We would like to bring to your notice that we have submitted the rebuttal and updated our draft following the reviews and comments. We have incorporated the reviewer comments in the updated draft. With the extended discussion period, we are glad to answer further questions. We again thank the reviewer for the constructive feedback. If our clarifications address your concerns, we request you to consider improving the score.

---

### Official Review · Reviewer_sdiR · 2024-11-09

**Soundness:** 3
**Presentation:** 2
**Contribution:** 3
**Rating:** 6
**Confidence:** 3

**Summary:**

I am not sure whether I completely understood the paper. It seems that the paper claims that the samples from an LLM are driven by two factors:

1. The underlying data distribution used to train LLM (or maybe data used to prompt them?).
2. Some societal judgment (coined as ideal) of the structure of the data.

For example, if the distribution of workout hours is mean = 7.5 but somehow if there is also a piece of information that says 7.5 hours is not sufficient/healthy, then the output of LLM related to workout hours would be larger than 7.5 hours.

----
I read the rebuttal and bought the implication of the result. My concerns on writing (more specifically, I felt the authors could bake something better than this with more time) remains though.

Make score converge after discussion.

**Strengths:**

The authors provided a quite interesting angle to explain how LLM sampling works. I also think the authors did some sufficiently rigorous assessment to validate their claims. Then I believe they also made a point that this judgmental component should be taken seriously because they “distorted” outputs of LLM from the data distribution.

**Weaknesses:**

I honestly feel it difficult to assess this paper. My hunch is this looks like a “fun fact” but reading the paper only left me with increased frustration — I dont feel I understand LLMs better.

But to be fair, this is conceptually quite new; I think for most ML but non-LLM specialized people, the intelligence offered by LLMs remain mysterious, and perhaps more disturbingly we dont have streamlined tools to make sense of these intelligence so providing stand-alone observations like this one perhaps is important.


I also feel the authors probably can find better ways to pitch that out over time. The paper looks written in a rush, e.g., first 3 pages are highly repetitive. The real important “meat” seems to be Sec. 3.1 to 3.3, which I read a few times before I feel I understand what it says. Some statements remain vague. For example, “the LLM is provided with N samples from this distribution as values associated with C”, are these N samples provided in the form of prompting, or part of training/fine tuning data.

**Questions:**

Did I understand correctly that the authors is pitching one stand-alone observation and do not claim they can explain why this happens (or have I missed other important messages)?

---

> ### Author Response · Authors · 2024-11-21
>
> We sincerely thank the reviewer for their honest and constructive feedback.
>
> > Clarifying implications of our proposed theory for better understanding LLMs
>
> We also view LLMs as a new form of intelligence (Embers of Autoregression: Understanding Large Language Models, reference: https://arxiv.org/abs/2309.13638,) and use grounded methodologies and principles from cognitive science to derive a theory which explains the heuristics behind LLM sampling its options. The topic of our study is inspired from the experiments done in possibility sampling in humans (refer: How we know what not to think? https://www.cell.com/trends/cognitive-sciences/fulltext/S1364-6613(19)30231-1,
> What comes to mind? https://www.sciencedirect.com/science/article/abs/pii/S0010027719302306) where it was investigated how indeed do humans narrow down options from the vast set of possibilities, most of which go unrealised. For example: If a car breaks down in the middle of the road, an agent faces innumerable decision-making options. It could either call for roadside assistance or a tow truck service on one end or attempt to push the car off the road and walk to the destination at the other extreme. It is difficult to consider/deliberate on all possible options. To narrow these options to a `consideration set’ within a computationally reasonable time, the agent needs to employ a heuristic driven approach (As discussed in the cognitive science papers referenced above).
> We investigate this thoroughly in LLMs and conclude that the heuristics used by LLM converges with that of humans. We observe that, both these decision making systems use a heuristic of both descriptive and prescriptive norms. We believe this convergence (and divergence in the exact prescriptive norms) uncovers the nature of decision making of LLMs, as a similar framework was used to explain the heuristics behind decision making by humans. Evidence that LLMs process information in ways resembling human heuristic reasoning we believe is not just a fun fact.
> Furthermore, the proposed theory has immediate practical implications like the medical case study presented in the paper. Besides multiple similar use cases, the theory would affect the next wave of synthetic data generation. This strengthens the claim that the paper has actionable insights.
>
>
> > Claim on explanation the observation:
>
> We believe our work improves the understanding of heuristics that drive LLM decision making. We acknowledge it is difficult to ‘explain why this happens ’, but since the heuristics  are shown to converge with humans we propose some hypotheses in the text.
> `One of the basic characteristics of System-1 is that it represents categories as norms or prototypical examples. System-1 has a model of the world that represents what is normal’ (Thinking fast and slow Daniel Kahneman). Also LLMs like neural networks have a feature representation for concepts/tokens from which drives it outputs.
> In humans prototypes tend to have an ideal component. Expanding on L221 of main text, a Robin might be considered a prototype of the category Bird over a penguin, for a prototypical bird is not a flightless bird! Flying, expected of birds (value), makes Robin a representative example of the group (Smith & Medin, 1981) over a penguin. Robin shares many common features with most birds also with high occurrence. Prototypes in humans are driven by both statistical occurance and ideal qualities.
> We show initial investigation to how prototypicality in LLMs have the same heuristics as humans. We believe more follow up research could build on the findings of this work as to explain the heuristics in more details.
> Furthermore there are papers that use LLM zero shot for applications like tabular data generation[1] where LLM is actually asked to produce more numbers that follow distribution of values in each feature. The implication of sampling bias even in the critical experiment designed to validate the phenomenon is practical.But this paper provides a theoretical framework based on human psychology and is not an application work. To address your comment we enhance the discussion on how this theory can guide future research or practical interventions for LLM-based systems.
>
> [1] Large Language Models(LLMs) on Tabular Data: Prediction, Generation, and Understanding - A Survey. Fang et al

---

> ### Author Response · Authors · 2024-11-21
>
> > Presentation of the paper: restructuring initial sections
>
> We acknowledge that the introductory sections may come across as repetitive. Our intent was to ensure that readers unfamiliar with LLMs and cognitive frameworks could fully understand the context before delving into our theory and experiments. However, we recognize the need for brevity and will streamline these sections in the revision. Please find the introduction and related work sections revised in the updated submission.
> We have defined sampling LLMs in para 2. Following reviewer suggestions we added this paragraph to give definition upfront in the main text and establish all the definitions used in the paper there. We request you to revisit the initial sections of our revised draft. Between the submission and now the language and presentation has been worked on, specifically we have restructured the Introduction, discussion and related work.
>
> > These N samples provided in the form of prompting, or part of training/fine tuning data?
>
> All evaluations in the paper are on zero shot behavior of LLMs. Hence all input is given by prompt. We acknowledge the confusion and clarify this in the paper.
> We agree that the current work does not delve deeply into the root causes of the observed prescriptive tendencies, and we pitch this an initial evaluation. But we believe these structured investigations that uncover the heuristics of LLMs can help us understand these systems better. Our goal is to propose the theory and validate it empirically, which we believe is a systematic direction much needed to understanding the mechanism of LLM.
> We again thank the reviewer for their honest feedback. If our clarifications address your concerns, we request you to consider improving the score.

---

### Note · Authors · 2024-12-15

I have read and agree with the venue's withdrawal policy on behalf of myself and my co-authors.